# MissDiff: Training Diffusion Models on Tabular Data with Missing Values

## Abstract

The diffusion model has shown remarkable performance in modeling data distributions and synthesizing data. However, the vanilla diffusion model requires complete or fully observed training data. Incomplete data is a common issue in various real-world applications, including healthcare and finance, particularly when dealing with tabular datasets. This work considers learning from data with missing values and generating synthetic complete data, beyond missing value imputations. The main challenge for this setting is that two-stage inference frameworks, the "impute-then-generate" pipeline or the "generate-then-impute" pipeline, are either biased or computationally expensive. To address this challenge, we present a unified and principled diffusion-based framework. Our method models the score of complete data distribution by denoising score matching on data with missing values. We prove that the proposed method can recover the score of the complete data distribution, and the proposed training objective serves as an upper bound for the negative likelihood of observed data. In the presence of incomplete training data, the proposed method can be used for synthetic data generation, as well as missing value imputations based on the learned generative model. Extensive experiments on imputation tasks together with generation tasks demonstrate that our proposed framework outperforms existing state-of-the-art approaches on multiple tabular datasets.

## 1 Introduction

Diffusion models have emerged as an effective tool for modeling the data distribution and synthesize various types of data, such as images (Ho et al., 2020; Song et al., 2021b; Dhariwal & Nichol, 2021; Rombach et al., 2021), videos (Ho et al., 2022), point clouds (Luo & Hu, 2021), and tabular data (Kim et al., 2023; Kotelnikov et al., 2022). These machine learning models typically rely on high-quality training data, which are usually expected to be free of missing values. In reality, it is often challenging to obtain complete data, particularly in healthcare, finance, recommendation systems, and social networks, due to privacy concerns, high cost or sampling difficulties, and the skewed distribution of user-generated content. For example, the respiratory rate of a patient may not have been measured, either because it was deemed unnecessary or was accidentally not recorded (Yoon et al., 2017; Alaa et al., 2016; Yoon et al., 2018a). Additionally, some information may be difficult or even dangerous to acquire, such as information obtained through a biopsy, which may not have been gathered for those reasons (Yoon et al., 2018b).

Moreover, deep generative models, particularly diffusion models, can be used to augment training data and enhance the performance of image classification tasks (Azizi et al., 2023; You et al., 2023) and adversarial robustness (Gowal et al., 2021; Sehwag et al., 2022; Ouyang et al., 2022). Following this idea, we can achieve better performance for downstream tasks by utilizing generative model learning on incomplete data for synthetic data generation. Therefore, in this work, we focus on learning a generative model from training data containing missing values and synthesize *new complete data*, not just imputing the missing value.

Numerous studies have been proposed to deal with missing values in the training data. Some of them can be used to generate new complete samples. We divide these works into two branches. The first line of work follows the "impute-then-generate" framework, i.e., they first complete the data and then learn a generative model on imputed data. Some approaches involve deleting instances or

features with missing data or replacing missing values with the mean of observed values for that feature. Other methods employ machine learning approaches (van Buuren & Groothuis-Oudshoorn, 2011; Bertsimas et al., 2017) or deep generative models for imputation tasks (Yoon et al., 2018a; Biessmann et al., 2019; Wang et al., 2020; Ipsen et al., 2022; Muzellec et al., 2020). It has been shown that imputation may reduce the diversity of the training data and may lead to biased performance in downstream tasks (Bertsimas et al., 2021; Ipsen et al., 2022).

Another line of work follows the "generate-then-impute" framework, i.e., they first learn the generative model directly on the data with missing values and generate synthetic samples that also have missing values, and then impute them. Some approaches use the variational lower bound on observed data to train a VAE-based model (Ipsen et al., 2021; Nazábal et al., 2018; Ma et al., 2020; Mattei & Frellsen, 2019; Valera et al., 2017). Other methods use adversarial training by optimizing a min-max objective to train a GAN-based model (Yoon et al., 2018a; Li et al., 2019; Li & Marlin, 2020). All of these works except Ipsen et al. (2021) require further assumptions on missing mechanisms.

Most importantly, all of the works except Li et al. (2019); Li & Marlin (2020) need two-stage inference for synthesizing new complete samples, while Li et al. (2019); Li & Marlin (2020) require training additional networks and thereby increase the computational costs[1]. In this work, we propose a unified framework, which we call *MissDiff*, for both imputation and synthetic complete data generation. Specifically, *MissDiff* can perform imputation tasks together with generation tasks without two-stage inference or training additional neural networks. *MissDiff* models the score (gradient log density) of complete data distribution by denoising score matching on data with missing values. We present the theoretical justification of *MissDiff* on recovering the oracle score function of the complete data and also upper bounding the negative likelihood of the observed data under mild assumptions.

We primarily utilize *tabular* data for the numerical experiments, as tabular data is a commonly encountered data type and frequently contains missing values in various applications Yoon et al. (2017); Alaa et al. (2016). Moreover, by considering tabular data as an example, we simultaneously study the missing value scenarios in categorical and continuous variables, which are both contained in tabular type data.

To verify the effectiveness of *MissDiff*, we conduct a suite of numerical experiments under various missing mechanisms. For both imputation tasks and generation tasks, *MissDiff* outperforms existing state-of-the-art methods in most settings by a considerable margin.

Our contributions can be summarized as follows.

- We propose a unified framework, which we call *MissDiff*, for imputation and synthetic complete samples generation by learning from data with missing values.
- We provide the theoretical justifications of *MissDiff* on recovering the oracle score function of the complete data and upper bounding the negative likelihood of the observed data under mild assumptions.
- *MissDiff* outperforms existing state-of-the-art methods in most settings on both imputation tasks and generation tasks on multiple real tabular datasets under different missing mechanisms.

The rest of the paper is organized as follows. Section 2 reviews the setup of the missing data mechanism and the score-based generative model. Section 3 introduces the proposed method and theoretically characterizes the effectiveness of the proposed method. Numerical results are given in Section 4. We conclude the paper in Section 5. All proofs and additional numerical experiments are deferred to the appendix.

## 1.1 RELATED WORK

**Learning from data with missing value:** Numerous studies have been proposed to deal with missing values in the training data. Variational Autoencoder (VAE) based models (Ipsen et al., 2021; Nazábal et al., 2018; Ma et al., 2020; Mattei & Frellsen, 2019; Valera et al., 2017; Ipsen et al., 2022) maximize the evidence low bound of the observed data, while Generative Adversarial Network (GAN) based models (Yoon et al., 2018a; Li et al., 2019; Li & Marlin, 2020) employ adversarial training

---

[1]More details can be found in Section 1.1.

for both the generative and discriminative models. Recently, Tashiro et al. (2021) proposes the conditional score-based generative model for time series imputation and Zheng & Charoenphakdee (2022) adopts the conditional score-based diffusion model proposed in Tashiro et al. (2021) for imputing tabular data. However, all of the above works mainly focus on imputation tasks. They either need two-stage inference, such as learning a generative model on imputed data or imputing the generated data containing missing values, or require training additional networks[2]. For example, Li et al. (2019) trains two generator-discriminator pairs for the masks and data respectively, which increases the computational cost, and Li & Marlin (2020) adopts Partial Bidirectional GAN, which requires an encoding and decoding network for the generator. Moreover, Nazábal et al. (2018); Ma et al. (2020) require training a different VAE independently to each data dimension.*MissDiff* is a unified framework for imputation and generation tasks without two-stage inference or training additional networks.

**Generative model for tabular data:**   Tabular data, as mixed-type data that typically contains both categorical and continuous variables, has attracted significant attention in the field of machine learning. The presence of mixed variable types and class imbalance for discrete variables make it a challenging task to model tabular data. Recently, several deep learning models have been proposed for tabular data generation (Xu et al., 2019; Choi et al., 2017; Srivastava et al., 2017; Park et al., 2018; Kim et al., 2021; Finlay et al., 2020; Kim et al., 2023; Kotelnikov et al., 2022). Among these methods, (Kotelnikov et al., 2022) employs Gaussian transitions for continuous variables and multinomial transitions for discrete variables, while (Kim et al., 2023) proposes a self-paced learning technique and a fine-tuning strategy for score-based models and achieves state-of-the-art performance in tabular data generation. Moreover, the discrete Score Matching methods proposed in Meng et al. (2022) and Sun et al. (2023) can also be employed to handle discrete variables in tabular data. However, all of the methods mentioned above did not take missing values in the training data into consideration.

## 2  PROBLEM SETUP AND PRELIMINARIES

### 2.1  TRAINING WITH MISSING DATA

We aim to learn a diffusion-based generative model from training data that may contain a certain proportion of missing values. Following the settings in Little & Rubin (1988); Li et al. (2019); Ipsen et al. (2022), we denote the underlying complete $d$-dimensional data as $\mathbf{x} = (x_1, \ldots, x_d) \in \mathcal{X}$ and assume it is sampled from the unknown true data-generating distribution $p_0(\mathbf{x})$. Here, each variable $x_i$, $i = 1, \ldots, d$, can be either categorical or continuous. For each data point $\mathbf{x}$, suppose there is a binary mask $\mathbf{m} = (m_1, \ldots, m_d) \in \{0, 1\}^d$ which indicates the missing entry for the current sample, i.e.,

$$m_i = \begin{cases} 1 & \text{if } x_i \text{ is observed,} \\ 0 & \text{if } x_i \text{ is missing.} \end{cases}$$

Then, the observed (incomplete) data $\mathbf{x}^{\text{obs}} = \mathbf{x} \odot \mathbf{m} + \text{na} \odot (\mathbf{1} - \mathbf{m})$, where na indicates the missing value, $\odot$ denotes element-wise multiplication, and $\mathbf{1}$ is the all-one vector.

Suppose we have $n$ complete (unobservable) data points $\mathbf{x}_1, \ldots, \mathbf{x}_n \overset{iid}{\sim} p_0(\mathbf{x})$ and simultaneously $n$ corresponding masks $\mathbf{m}_1, \ldots, \mathbf{m}_n$ generated from a specific missing data mechanism detailed later. Then, the observed data samples are denoted as $S^{\text{obs}} = \{\mathbf{x}_i^{\text{obs}}\}_{i=1}^n$. The missing mechanisms can be categorized based on the relationships between the mask $\mathbf{m}$ and the complete data $\mathbf{x}$ (Little & Rubin, 1988) as follows,

- Missing Completely At Random (MCAR): mask $\mathbf{m}$ is independent from complete data $\mathbf{x}$.
- Missing At Random (MAR): mask $\mathbf{m}$ only depends on the observed value $\mathbf{x}^{\text{obs}}$.
- Not Missing At Random (NMAR): $\mathbf{m}$ depends on the observed value $\mathbf{x}^{\text{obs}}$ and missing value.

Compared with previous work which typically develops their algorithms and theoretical foundations under the M(C)AR assumption Li et al. (2019); Ipsen et al. (2022); Yoon et al. (2018a); Li & Marlin

---

[2]Additional network means the extra network needed compares with the same model dealing with complete data.

(2020); Mattei & Frellsen (2019), our method and theoretical guarantees aim to provide a general framework for learning on incomplete data and generate complete data. By modeling the score of the complete data distribution from the observed data, we only require mild assumptions of missing mechanisms for recovering the oracle score (we refer to Theorem 3.2). In the following, we provide a brief introduction to the score-based generative model.

## 2.2 SCORE-BASED GENERATIVE MODEL

In this work, we adopt the diffusion model[3] as the prototype for developing our proposed method. We propose to train the model with missing values directly without the need for prior imputation. We first briefly review the key components of score-based generative models (Ho et al., 2020; Song et al., 2021b).

Score-based generative models are a class of generative models that learn the score function, which is the gradient of the log density of the data distribution. These models have gained attention due to their flexibility and effectiveness in capturing complex data distributions. Following the notation in Song et al. (2021b), the score-based generative models are based on a forward stochastic differential equation (SDE), $\mathbf{x}(t)$ with $t \in [0, T]$, defined as (which corresponds to Eq (5) in Song et al. (2021b))

$$\mathrm{d}\mathbf{x}(t) = \mathbf{f}(\mathbf{x}(t), t)\mathrm{d}t + g(t)\mathrm{d}\mathbf{w}, \tag{1}$$

where $\mathbf{w}$ is the standard Wiener process (Brownian motion), $\mathbf{f}(\cdot, t) : \mathbb{R}^d \to \mathbb{R}^d$ is a vector-valued function called the drift coefficient of $\mathbf{x}(t)$, and $g(\cdot) : \mathbb{R} \to \mathbb{R}$ is a scalar function known as the diffusion coefficient of $\mathbf{x}(t)$.

The solution of a stochastic differential equation is a continuous trajectory of random variables $\{\mathbf{x}(t)\}_{t \in [0, T]}$. Let $p(\mathbf{x})$ denote the path measure for the trajectory $\mathbf{x}$ on $[0, T]$, $p_t(\mathbf{x})$ denote the marginal probability density function of $\mathbf{x}(t)$, and $p(\mathbf{x}(t)|\mathbf{x}(s))$ denote the conditional probability density of $\mathbf{x}(t)$ conditioned on $\mathbf{x}(s)$, where $s < t$ is a previous time point. When constructing the SDE, we let $p_0(\mathbf{x})$ be the true data distribution, and after perturbing the data according to the SDE, the data distribution becomes $p_T(\mathbf{x})$ which is close to a tractable noise distribution, usually set as the standard Gaussian distribution.

The data generation process is performed via the reverse SDE, i.e., first sampling data $\mathbf{x}_T$ from $p_T(\mathbf{x})$ and then generate $\mathbf{x}_0$ through the reverse of (1). For any SDE in (1), the corresponding backward/reverse process is as follows (we refer Anderson (1982) for detailed explanation):

$$\mathrm{d}\mathbf{x}(t) = \left[\mathbf{f}(\mathbf{x}(t), t) - g(t)^2 \nabla_{\mathbf{x}} \log p_t(\mathbf{x})\right] \mathrm{d}t + g(t)\mathrm{d}\overline{\mathbf{w}}, \tag{2}$$

where $\overline{\mathbf{w}}$ is a standard Wiener process when time flows backwards from $T$ to 0, and $\mathrm{d}t$ is an infinitesimal negative time step.

We can generate new data by running backward the reverse-time SDE (2) when the score of each marginal distribution, $\nabla_{\mathbf{x}} \log p_t(\mathbf{x})$ is known. Score Matching (Hyvärinen, 2005; Vincent, 2011; Song et al., 2019) can be used for training a score-based model $\mathbf{s}_{\boldsymbol{\theta}}(\mathbf{x}(t), t)$ to estimate the score:

$$\boldsymbol{\theta}^* = \arg\min_{\boldsymbol{\theta}} \mathbb{E}_t \left\{ \lambda(t) \mathbb{E}_{p(\mathbf{x}(0))} \mathbb{E}_{\mathbf{x}(t)|\mathbf{x}(0)} \left[ \left\| \mathbf{s}_{\boldsymbol{\theta}}(\mathbf{x}(t), t) - \nabla_{\mathbf{x}(t)} \log p(\mathbf{x}(t)|\mathbf{x}(0)) \right\|_2^2 \right] \right\}, \tag{3}$$

where $\lambda : [0, T] \to \mathbb{R}_{>0}$ is a positive weighting function, $t$ is uniformly sampled over $[0, T]$, $\mathbf{x}(0) \sim p_0(\mathbf{x})$ and $\mathbf{x}(t) \sim p(\mathbf{x}(t)|\mathbf{x}(0))$. The local consistency of score matching is shown in (Hyvärinen, 2005), i.e., $\mathbb{E}_{p(\mathbf{x}(0))}[\|\mathbf{s}_{\boldsymbol{\theta}}(\mathbf{x}) - \nabla_{\mathbf{x}} \log p(\mathbf{x})\|_2^2] = 0 \Leftrightarrow \boldsymbol{\theta} = \boldsymbol{\theta}^*$ under the assumption that there exists an unique $\boldsymbol{\theta}^*$ such that the true score function $\nabla_{\mathbf{x}} \log p(\mathbf{x})$ can be represented by $s_{\boldsymbol{\theta}^*}$. Vincent (2011) builds the connection between Denoising Score Matching and Score Matching, and Song et al. (2019) further proves Sliced Score Matching can learn the consistent estimator of the oracle score and the asymptotic normality for the Sliced Score Matching.

## 3 METHOD

In this section, we first discuss the room for improvement in two existing frameworks for synthesizing new complete data in section 3.1. Then, we propose a unified framework, *MissDiff*, for learning a

---

[3]We use diffusion model and score-based generative model interchangeably as the equivalence is built by Song et al. (2021b).

generative model from incomplete data in section 3.1. The theoretical guarantees of *MissDiff* are provided in section 3.3.

## 3.1 THE LIMITATION OF TWO EXISTING FRAMEWORKS

In general, there are two paradigms to learn a generative model from incomplete data. The first paradigm is "impute-then-generate" framework, i.e., constructing a complete training dataset first and then learning a generative model on the complete data. We can either delete instances (rows) or features (columns) with missing data or adopt traditional imputation methods or training machine learning imputation models (van Buuren & Groothuis-Oudshoorn, 2011; Bertsimas et al., 2017) or deep generative models for imputation tasks (Vincent et al., 2008; Yoon et al., 2018a; Biessmann et al., 2019; Wang et al., 2020; Ipsen et al., 2022; Muzellec et al., 2020). However, this pipeline may bring bias to the training objective. We clarify this claim in remark 3.1.

*Remark* 3.1 ("Impute-then-generate" framework is biased). Inspired by the analysis pipeline of "impute-then-regress" (Bertsimas et al., 2021; Ipsen et al., 2022) for the prediction task, we can study a corresponding framework for the generation task. The generative model $p_\phi$ represents the probability distribution of the synthetic data $\mathbf{x}$. Under the maximum likelihood framework, $\phi^* := \arg\max_\phi \mathbb{E}_{\mathbf{x} \sim p_0(\mathbf{x})}[\log p_\phi(\mathbf{x})]$. When data has missing values, the general approach, known as "impute-then-generate", may be used in practice. In this approach, the observed data $\mathbf{x}^{\mathrm{obs}}$ is first imputed using an imputation model $f_\varphi$, where $f_\varphi(\mathbf{x}^{\mathrm{obs}})$ is trained by the regression loss $\mathbb{E}_{(\mathbf{x}^{\mathrm{obs}}, \mathbf{x}^{\mathrm{miss}}) \sim p_0(\mathbf{x})} \|f_\varphi(\mathbf{x}^{\mathrm{obs}}) - \mathbf{x}^{\mathrm{miss}}\|^2$ with $\mathbf{x}^{\mathrm{miss}}$ as the ground truth value[4]. The optimal $f_\varphi^*(\mathbf{x}^{\mathrm{obs}})$ satisfies $f_\varphi^*(\mathbf{x}^{\mathrm{obs}}) = \mathbb{E}_{p_0(\mathbf{x}^{\mathrm{miss}}|\mathbf{x}^{\mathrm{obs}})}[\mathbf{x}^{\mathrm{miss}}])$. Then, the generative model is trained by maximizing the likelihood of imputed data, i.e., $\max_\phi \log p_\phi(\mathbf{x}^{\mathrm{obs}}, \mathbf{x}^{\mathrm{miss}} := f_\varphi(\mathbf{x}^{\mathrm{obs}}))$. In general, $\mathbb{E}_{p_0(\mathbf{x}^{\mathrm{miss}}|\mathbf{x}^{\mathrm{obs}})}[p_\phi(\mathbf{x}^{\mathrm{obs}}, \mathbf{x}^{\mathrm{miss}})] \neq p_\phi(\mathbf{x}^{\mathrm{obs}}, \mathbb{E}_{p_0(\mathbf{x}^{\mathrm{miss}}|\mathbf{x}^{\mathrm{obs}})}[\mathbf{x}^{\mathrm{miss}}])$. Therefore, this pipeline is biased because the optimal single imputation can no longer capture the data variability.

The "Generate-then-impute" framework could also have some drawbacks, e.g., they need MCAR or MAR condition for the missing mechanism or require training additional networks[5]. Moreover, both frameworks require two-stage inference for generating new complete data, which highlights the need for alternative unbiased and unified approaches that can handle missing data directly and more effectively.

In this work, we show that modeling the score of the complete data distribution can help to form a unified way for both imputation and generation tasks. However, the diffusion model mentioned in Section 2.2 is unable to directly deal with data with missing values. Therefore, we propose a diffusion-based framework designed for training diffusion models on tabular data with missing values, which improves the above limitations.

## 3.2 *MissDiff*: DENOISING SCORE MATCHING ON MISSING DATA

We propose the following Denoising Score Matching method for data with missing values. Instead of using Eq (3) for learning the score-based model $\mathbf{s}_\theta(\mathbf{x}(t), t)$, we propose *MissDiff* as solution to

$$\boldsymbol{\theta}^* = \arg\min_{\boldsymbol{\theta}} J_{DSM}(\boldsymbol{\theta}) := \frac{T}{2} \mathbb{E}_t \Big\{ \lambda(t) \mathbb{E}_{\mathbf{x}^{\mathrm{obs}}(0)} \mathbb{E}_{\mathbf{x}^{\mathrm{obs}}(t)|\mathbf{x}^{\mathrm{obs}}(0)}$$
$$\Big[ \big\| \big( \mathbf{s}_{\boldsymbol{\theta}}(\mathbf{x}^{\mathrm{obs}}(t), t) - \nabla_{\mathbf{x}^{\mathrm{obs}}(t)} \log p(\mathbf{x}^{\mathrm{obs}}(t) \mid \mathbf{x}^{\mathrm{obs}}(0)) \big) \odot \mathbf{m} \big\|_2^2 \Big] \Big\}, \quad (4)$$

where $\lambda(t)$ is a positive weighting function, $\mathbf{m} = \mathbb{1}\{\mathbf{x}^{\mathrm{obs}}(0) \neq \mathrm{na}\}$ indicated the observed entries in $\mathbf{x}^{\mathrm{obs}}$ and $p(\mathbf{x}^{\mathrm{obs}}(t)|\mathbf{x}^{\mathrm{obs}}(0)) = \mathcal{N}(\mathbf{x}^{\mathrm{obs}}(t); \mathbf{x}^{\mathrm{obs}}(0), \beta_t \mathbb{I})$ is the Gaussian transition kernel. More implementation details can be found in Appendix B.4.

More specifically, we mainly adopt the Variance Preserving (VP) SDE in this paper although Variance Exploding (VE) SDE (Song et al., 2021b) is also applicable. The forward diffusion process of the Variance Preserving SDE is defined as (which corresponds to Eq (11) in (Song et al., 2021b)):

$$\mathrm{d}\mathbf{x} = -\frac{1}{2}\beta(t)\mathbf{x}\mathrm{d}t + \sqrt{\beta(t)}\mathrm{d}\mathbf{w},$$

---

[4] Here the notation $(\mathbf{x}^{\mathrm{obs}}, \mathbf{x}^{\mathrm{miss}})$ means the complete data $\mathbf{x}$.

[5] More details can be found in Section 1.1.

where $\{\beta_t \in (0,1)\}_{t \in (0,T)}$ is the increasing sequence denoting the variance schedule. Algorithm 1 demonstrates the Denoising Score Matching objective on missing data[6]. As long as the score function of complete data distribution is learned, we can adopt Algorithm 2 for imputation and Algorithm 3 for generating complete samples, which are provided in the Appendix B.3.

---

**Algorithm 1** *MissDiff*: Denoising Score Matching on Data with Missing Values

---

**Require:** Diffusion process hyperparameter $\beta_t, \sigma_t$, denote $\alpha_t = 1 - \beta_t$ and $\bar{\alpha}_t = \prod_{s=1}^{t} \alpha_s$.
1: **repeat**
2:    Sample $\mathbf{x}_0^{\text{obs}}$ according to the data distribution and missing mechanism;
3:    Infer mask $\mathbf{m} = \mathbb{1}[\mathbf{x}_0^{\text{obs}} \neq \text{na}]$;
4:    $t \sim \text{Uniform}(\{1, \ldots, T\})$;
5:    $\epsilon_t \sim \mathcal{N}(\mathbf{0}, \mathbf{I})$;
6:    Take gradient descent step on

$$\nabla_\theta \left\| (\epsilon_t - \mathbf{s}_{\boldsymbol{\theta}}(\sqrt{\bar{\alpha}_t}\mathbf{x}_0^{\text{obs}} + \sqrt{1 - \bar{\alpha}_t}\epsilon_t, t)) \odot \mathbf{m} \right\|^2 .$$

7: **until** converged.

---

## 3.3 THEORETICAL GUARANTEES OF *MissDiff*

In this section, we examine the effectiveness of *MissDiff* by theoretically characterizing the Score Matching objective under mild conditions on the missing mechanisms and build a further connection between Score Matching and maximizing likelihood objective for training the diffusion model.

In the following theorem, we present our first theoretical result that verifies that Denoising Score Matching on missing data can learn the oracle score, i.e, the score on complete data. Theorem 3.2 states that the global optimal solution of Denoising Score Matching on missing data obtained by *MissDiff* is the same as the oracle score, as long as we do not have a variable that is completely missing in the training data. The proof can be found in Appendix A.1.

**Theorem 3.2.** *Denote $\rho_i$, $i \in \{1, 2, ..., d\}$ as the percentage of missing samples for the $i$-th entry in the training data. Suppose $\rho_{max} := \max_{i=1,...,d} \rho_i < 1$. Let $\boldsymbol{\theta}^*$ be the solution to the training objective of MissDiff defined in Eq (4). Then we have*

$$\mathbf{s}_{\boldsymbol{\theta}^*}(\mathbf{x}(t), t) = \nabla_{\mathbf{x}(t)} \log p_t(\mathbf{x}(t)).$$

It is well known that with careful design of the weighting function $\lambda_t$, Denoising Score Matching can upper bound the negative log-likelihood of the diffusion model on the complete data (Song et al., 2021a). Therefore, it is straightforward to extend such a connection to incomplete data scenarios, which is detailed in the following theorem. These results provide insightful connections between the training objective of *MissDiff* and the maximum likelihood objective of the generative model on observed data.

**Theorem 3.3.** *The objective function of Denoising Score Matching on missing data is an upper bound for the negative likelihood of the generative model on observed data $\mathbf{x}^{obs}$ up to a constant, that is, for $\lambda_t = \beta_t$ and under the same condition of Theorem 3.2 and mild regularity conditions detailed in Appendix A.2, we have*

$$-\mathbb{E}_{p(\mathbf{x}^{obs})}\left[\log p_{\boldsymbol{\theta}}(\mathbf{x})\right] \leq \frac{1}{1 - \rho_{max}} J_{\text{DSM}}(\boldsymbol{\theta}) + C_1,$$

*where $C_1$ is a constant independent of $\boldsymbol{\theta}$.*

The proof of Theorem 3.3 can be found in Appendix A.2. When there are missing values, Theorem 3.3 shows that the Denoising score matching on incomplete data still upper bounds the likelihood of the incomplete data up to a constant coefficient $1/(1 - \rho_{\max})$. When there is no data missing, $\rho$ is all zero vector, then we have $1/(1 - \rho_{\max}) = 1$ and Theorem 3.3 degenerates to the Corollary 1 in Song et al. (2021a), i.e.,

$$-\mathbb{E}_{p(\mathbf{x})}[\log p_{\boldsymbol{\theta}}(\mathbf{x})] \leq J_{\text{DSM}}(\boldsymbol{\theta}; g(\cdot)^2) + C_1,$$

where the $J_{\text{DSM}}(\boldsymbol{\theta}; g(\cdot)^2)$ is the Denoising Score Matching objective on complete data.

---

[6]We write $\mathbf{x}(t)$ as $\mathbf{x}_t$ in the algorithm box for simplicity.

## 4 EXPERIMENTS

In this section, we demonstrate the effectiveness of the proposed *MissDiff* against existing state-of-the-art models. Since most of the approaches dealing with missing data work on imputation tasks, we compare with them in Section 4.1. Then, we mainly focus on the complete synthetic data generation task, which was much less evaluated in the literature with missing data. We present a careful experimental setup, including datasets, baseline models, and evaluation criterion, in Section 4.2. The detailed experimental results under different missing mechanisms are in Section 4.3.

### 4.1 IMPUTATION TASKS

**Datasets:** We follow the experimental setup as Zheng & Charoenphakdee (2022), that is evaluating *MissDiff* on six UCI Machine Learning Repository (Kelly et al.). The details of the dataset and the missing mechanism can be found in Appendix B.1.

**Baseline Methods:** We compare *MissDiff* with (i) the simple imputation method that uses mean values for continuous values and mode values for discrete variables (Mean / Mode), (ii) Multiple Imputation by Chained Equations (MICE) with linear regression (MICE_linear) (White et al., 2011), (iii) MICE based on random forest (MissForest) (Stekhoven, 2015), (iv) GAIN (Yoon et al., 2018a), (v) MIWAE (Mattei & Frellsen, 2019), and (vi) CSDI_T (Zheng & Charoenphakdee, 2022).

**Results:** The performance comparison of *MissDiff* with state-of-the-art imputation approaches is presented in Table 1. For most datasets, *MissDiff* achieves the lowest Root Mean Squared Error (RMSE).

Table 1: Evaluation on imputation tasks. The standard deviations of five independent trails are shown in the parenthesis. The *lower* the RMSE, the *better* the performance.

| Method | Census | Breast | Wine | Concrete | Libras | diabetes |
|---|---|---|---|---|---|---|
| Mean /Mode | 0.120(0.003) | 0.263(0.009) | 0.076(0.003) | 0.217(0.007) | 0.099(0.001) | 0.222(0.003) |
| MICE(linear) | 0.101(0.002) | 0.154(0.011) | 0.065(0.003) | 0.153(0.006) | 0.034(0.001) | 0.263(0.002) |
| MissForest | 0.112(0.004) | 0.163(0.014) | 0.060(0.002) | 0.173(0.005) | 0.024(0.001) | 0.216(0.003) |
| GAIN | 0.123(0.057) | 0.165(0.006) | 0.072(0.004) | 0.203(0.007) | 0.089(0.006) | 0.202(0.003) |
| MIWAE | 0.113(0.042) | 0.1874(0.079) | 0.074 (0.005) | 0.195(0.006) | 0.083(0.003) | 0.194(0.081) |
| CSDI_T | 0.099(0.003) | 0.153(0.003) | 0.065(0.004) | **0.131**(**0.008**) | **0.011**(**0.001**) | 0.197(0.001) |
| *MissDiff* | **0.089(0.006)** | **0.136(0.002)** | **0.053(0.001)** | 0.161(0.001) | 0.0787(0.002) | **0.051(0.004)** |

### 4.2 EXPERIMENTAL SETUP FOR GENERATION TASK

**Datasets:** We present a suite of numerical evaluations of the proposed *MissDiff* approach on a simulated Bayesian Network data, a real Census tabular dataset (Kohavi & Becker, 1996), and the MIMIC4ED tabular dataset (Xie et al., 2022), with various proportions of missing values. The details of the dataset and the missing mechanism can be found in Appendix B.2.

**Baseline Methods:** Since few previous works provide the experimental results of the generative models learned on tabular data with missing values for generating new complete samples, we mainly compare with the following four baseline methods:

1. *Diff-delete*: Learn a vanilla diffusion model after deleting rows containing missing values.

2. *Diff-mean*: Learn a vanilla diffusion model after imputing missing values using the mean value in that column.

3. STaSy (Kim et al., 2023) with the above two data completion methods. STaSy is the state-of-the-art diffusion model on tabular data, which outperforms MedGAN (Choi et al., 2017), VEEGAN (Srivastava et al., 2017), CTGAN (Xu et al., 2019), TVAE (Xu et al., 2019), TableGAN (Park et al., 2018), OCTGAN (Kim et al., 2021), RNODE (Finlay et al., 2020) by a large margin.

We use the variance-preserving SDE with the time duration $T = 100$ for the Bayesian Network and Census dataset and $T = 150$ for the MIMIC4ED dataset. We use the standard pre/post-processing of tabular data to deal with mixed-type data (Kim et al., 2023; Kotelnikov et al., 2022; Zheng & Charoenphakdee, 2022), i.e., we use the min-max normalization for the continuous variables and reverse its scaler when generation. We use one-hot embedding for the discrete variables and use the rounding function after the softmax function when generation. We train the diffusion model for 250 epochs with batch size 64. For more details, please refer to Appendix B.4.

**Evaluation Criterion:** Following Xu et al. (2019); Kim et al. (2023); Kotelnikov et al. (2022), we use two types of criteria, *fidelity* and *utility*, to evaluate the quality of the synthetic data generated. To evaluate the *fidelity* of synthetic data compared with real data, we adopt a model-agnostic library, SDMetrics (Dat, 2023). The result is a float number range from 0 to 100%. The larger the score, the better the overall quality of synthetic data is.

To evaluate the *utility* of synthetic data, we follow the same pipeline of Kim et al. (2023), i.e., training various models, including Decision Tree, AdaBoost, Logistic/Linear Regression, MLP classifier/regressor, RandomForest, and XGBoost, on synthetic data, and validate the model on original training data, and test them with real test data. For classification tasks, we mainly use classification accuracy and also report AUROC, F1, and Weighted-F1 in Appendix B.5. For regression tasks, we mainly use RMSE and also report $R^2$ in the Appendix B.5. All the experiments are obtained from 3 repetitions.

### 4.3 EXPERIMENT RESULTS FOR GENERATION TASK

#### 4.3.1 SIMULATION STUDY

*Q1: How does MissDiff perform on different missing ratios against the vanilla diffusion model learned on the data completed by two baseline methods mentioned in section 4.2?*

Figure 1 summarizes the SDMetrics score on the simulated Bayesian Network dataset example. With the same diffusion model architecture and the same training hyperparameter, *MissDiff* achieves consistently better results against the vanilla diffusion model deleting the incomplete row or using the mean value for imputation when the missing ratio varies from 0.1 to 0.9. Moreover, the advantage of *MissDiff* becomes more obvious for large missing ratios. These experimental results verify the motivation of *MissDiff* proposed in Remark 3.1 that the learning objective of impute-then-generate is biased. Directly learning on the missing data can significantly enhance the performance of the learned generative model.

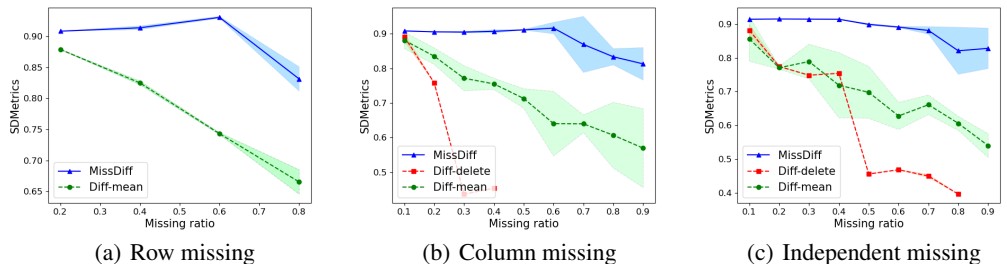

(a) Row missing      (b) Column missing      (c) Independent missing

Figure 1: *Fidelity* evaluation of *MissDiff* on data generated by Bayesian Network under different missing ratios. We shade the area between mean $\pm$ std.

#### 4.3.2 REAL TABULAR DATASETS

*Q2: How does MissDiff perform on more complicated real-world data and compared with state-of-the-art generative model on tabular data?*

Table 2 demonstrates the effectiveness of *MissDiff* on the Census dataset under MCAR. STaSy is a state-of-the-art generative model for tabular data, which means *MissDiff* achieves quite good

performance on learning from incomplete data and generating complete data. More importantly, *MissDiff* achieves better performance than *STaSy-delete* and *STaSy-mean* even without adopting the self-paced learning technique and the fine-tuning strategy used by STaSy. More experiments and discussions can be found in Appendix B.5.

Table 2: *Utility* (classification accuracy) evaluation of *MissDiff* on Census dataset. "-" denotes the corresponding method cannot applied since no data $\mathbf{x}_i$ will be left after deleting the incomplete data. The *larger* the accuracy, the *better* the performance.

|  | *MissDiff* | *Diff-delete* | *Diff-mean* | *STaSy-delete* | *STaSy-mean* | CSDI_T |
|---|---|---|---|---|---|---|
| Row Missing | **79.48**% | - | 78.45% | - | 70.79% | 79.15% |
| Column Missing | 71.68% | 72.89% | 79.60% | 68.96% | 74.47% | **80.31%** |
| Independent Missing | **79.49**% | 75.39% | 75.96% | 78.36% | 77.34% | 79.12% |

*Q3: How does MissDiff perform on real application of large-scale Electronic Health Records data?*

Table 3 shows the performance of *MissDiff* on the MIMIC4ED dataset under MCAR. On this large dataset with dozens of continuous and discrete variables, *MissDiff* gives consistently better performance with the same training epochs (250 epochs).

Table 3: *Utility* (RMSE) evaluation of *MissDiff* on MIMIC4ED dataset. *Diff-delete* and *STaSy-delete* cannot be applied since no data $\mathbf{x}_i$ will be left after deleting the incomplete data. The *lower* the RMSE, the *better* the performance.

|  | *MissDiff* | *Diff-mean* | *STaSy-mean* |
|---|---|---|---|
| Row Missing | **1.826** | 2.166 | 1.894 |
| Column Missing | **1.834** | 2.011 | 1.935 |
| Independent Missing | **1.852** | 2.483 | 1.972 |

*Q4: How does MissDiff perform on other missing mechanisms beyond MCAR, i.e., MAR and NMAR?*

Table 4 demonstrates the effectiveness of *MissDiff* on the Census dataset beyond MCAR. The results show the great potential of learning directly on the missing data when the missing mechanism is not MCAR, which cannot be easily dealt with by previous methods (Li et al., 2019; Ipsen et al., 2022; Yoon et al., 2018a; Li & Marlin, 2020).

Table 4: *Utility* (classification accuracy) evaluation of *MissDiff* on Census dataset under MAR, NMAR. The *larger* the accuracy, the *better* the performance.

|  | *MissDiff* | Diff-delete | Diff-mean | *STaSy-delete* | *STaSy-mean* | CSDI_T |
|---|---|---|---|---|---|---|
| MAR | **79.95**% | 69.48% | 77.43% | 71.28% | 73.65% | 79.42% |
| NMAR | **80.95**% | 66.50% | 80.03% | 78.11% | 73.92% | 80.23% |

## 5 CONCLUSION AND DISCUSSION

We propose a diffusion-based generative framework, called *MissDiff*, for synthetic data generation trained on data with missing values. Compared with the two-stage inference pipeline, "impute-then-generate" framework or "generate-then-impute" framework, *MissDiff* is a unified, unbiased, and computationally friendly framework. The theoretical justification for *MissDiff*'s effectiveness is provided. Moreover, extensive numerical experiments demonstrate strong empirical evidence for the effectiveness of *MissDiff*.

**Limitations and broader impact:** Overall, this research presents a promising direction for handling missing data in generative model training. A potential limitation of this work is that it has only been empirically validated on tabular data. For future directions, it would be interesting to see how *MissDiff* performs empirically with more complicated data types such as video or language data.

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

## A    PROOFS FOR SECTION 4

### A.1    PROOF OF THEOREM 3.2

In order to show Theorem 3.2, we aim to show that the optimal solution $\boldsymbol{\theta}^*$, which minimizes the objective function $J_{DSM}(\boldsymbol{\theta})$ satisfies $\mathbf{s}_{\boldsymbol{\theta}^*}(\mathbf{x}(t), t) = \nabla_{\mathbf{x}(t)} \log p_t(\mathbf{x}(t))$, i.e., the optimal solution to the population loss function can recover the oracle score function.

For the Gaussian transition distribution that we used with the isotropic covariance matrix, the score on the incomplete data is equivalent to the score on the complete data when performing element-wise multiplication with mask, i.e., $\nabla_{\mathbf{x}^{\mathrm{obs}}(t)} \log p(\mathbf{x}^{\mathrm{obs}}(t)|\mathbf{x}^{\mathrm{obs}}(0)) \odot \mathbf{m} = \nabla_{\mathbf{x}(t)} \log p(\mathbf{x}(t)|\mathbf{x}(0)) \odot \mathbf{m}$[7], where $\mathbf{m} = \mathbb{1}\{\mathbf{x}^{\mathrm{obs}}(0) \neq \mathrm{na}\}$ indicated the missing entries in $\mathbf{x}^{\mathrm{obs}}(0)$. Therefore, under certain conditions, we may first relate the Denosing Score Matching objective on missing data to the Denosing Score Matching objective on the complete data,

$$\mathbb{E}_{p(\mathbf{x}^{\mathrm{obs}}(0),\mathbf{m})} \mathbb{E}_{p(\mathbf{x}^{\mathrm{obs}}(t)|\mathbf{x}^{\mathrm{obs}}(0))} [\|(\mathbf{s}_{\boldsymbol{\theta}}(\mathbf{x}^{\mathrm{obs}}(t),t) - \nabla_{\mathbf{x}^{\mathrm{obs}}(t)} \log p(\mathbf{x}^{\mathrm{obs}}(t)|\mathbf{x}^{\mathrm{obs}}(0))) \odot \mathbf{m}\|_2^2]$$
$$= \mathbb{E}_{p(\mathbf{x}(0),\mathbf{m})} \mathbb{E}_{p(\mathbf{x}(t)|\mathbf{x}(0))} [\|(\mathbf{s}_{\boldsymbol{\theta}}(\mathbf{x}(t),t) - \nabla_{\mathbf{x}(t)} \log p(\mathbf{x}(t)|\mathbf{x}(0))) \odot \mathbf{m}\|_2^2].$$

Moreover, notice that we have

$$\mathbb{E}_{p(\mathbf{x}(0),\mathbf{m})} \mathbb{E}_{p(\mathbf{x}(t)|\mathbf{x}(0))} [\|(\mathbf{s}_{\boldsymbol{\theta}}(\mathbf{x}(t),t) - \nabla_{\mathbf{x}(t)} \log p(\mathbf{x}(t)|\mathbf{x}(0))) \odot \mathbf{m}\|_2^2]$$
$$= \mathbb{E}_{p(\mathbf{x}(\mathbf{0}),\mathbf{x}(\mathbf{t}))} \|(\mathbf{s}_{\boldsymbol{\theta}}(\mathbf{x}(t),t) - \nabla_{\mathbf{x}(t)} \log p_t(\mathbf{x}(t))) \odot \sqrt{\mathbb{E}_{p(\mathbf{m}|\mathbf{x}(0))}[\mathbf{m}]}\|_2^2,$$

where $\sqrt{z}$ denotes the element-wise operation on vector $z$. The last equation is because we take the conditional expectation of the binary mask $\mathbf{m}$ and since $\mathbf{m}_i \in \{0,1\}$ we have $\mathbb{E}[\mathbf{m}_i^2] = \mathbb{E}[\mathbf{m}_i]$ for any distribution of $\mathbf{m}$. Assuming that $\mathbb{E}_{p(\mathbf{m}|\mathbf{x}(0))}[\mathbf{m}] \equiv \mathbf{1} - \boldsymbol{\rho}$ with $\boldsymbol{\rho} = [\rho_1, \ldots, \rho_d]$ and $\rho_i < 1$, $i \in \{1, 2, ..., d\}$ being the population percentage of missing samples for the $i$-th entry, we have $\mathbb{E}_{p(\mathbf{m}|\mathbf{x}(0))}[\mathbf{m}] > 0$ and thus we can show the global optimal of Denoising Score Matching on missing data is the same as the oracle score.

### A.2    PROOF OF THEOREM 3.3

The notations are defined as follows. We let $\pi$ denote the pre-specified prior distribution (e.g., the standard normal distribution), $\mathcal{C}$ denote all continuous functions, and $\mathcal{C}^k$ denote the family of functions with continuous $k$-th order derivatives. Consider the MCAR missing mechanism. Denote $\rho_i, i \in \{1, 2, ..., d\}$ as the population percentage of missing samples for the $i$-th entry in the training data. Suppose $\max_{i=1,...,d} \rho_i < 1$. In addition, we make the same mild regularity assumptions as Song et al. (2021a) in the following.

**Assumption A.1.**    (i)  $p(\mathbf{x}) \in \mathcal{C}^2$ and $\mathbb{E}_{\mathbf{x} \sim p_0}[\|\mathbf{x}\|_2^2] < \infty$.

(ii)  $\pi(\mathbf{x}) \in \mathcal{C}^2$ and $\mathbb{E}_{\mathbf{x} \sim \pi}[\|\mathbf{x}\|_2^2] < \infty$.

(iii)  $\forall t \in [0, T] : f(\cdot, t) \in \mathcal{C}^1, \exists C > 0, \forall \mathbf{x} \in \mathbb{R}^d, t \in [0, T] : \|f(\mathbf{x}, t)\|_2 \leq C(1 + \|\mathbf{x}\|_2)$.

(iv)  $\exists C > 0, \forall \mathbf{x}, \mathbf{y} \in \mathbb{R}^d : \|f(\mathbf{x}, t) - f(\mathbf{y}, t)\|_2 \leq C\|\mathbf{x} - \mathbf{y}\|_2$.

(v)  $g \in \mathcal{C}$ and $\forall t \in [0, T], |g(t)| > 0$.

(vi)  For any open bounded set $\mathcal{O}, \int_0^T \int_{\mathcal{O}} \|p_t(\mathbf{x})\|_2^2 + dg(t)^2 \|\nabla_{\mathbf{x}} p_t(\mathbf{x})\|_2^2 \, \mathrm{d}\mathbf{x}\mathrm{d}t < \infty$.

---

[7]Assume $p(\mathbf{x}^{\mathrm{obs}}(t)|\mathbf{x}^{\mathrm{obs}}(0)) = \mathcal{N}(\mathbf{x}^{\mathrm{obs}}(t); \mu^{\mathrm{obs}}, \Sigma)$ and $p(\mathbf{x}(t)|\mathbf{x}(0)) = \mathcal{N}(\mathbf{x}(t); \mu, \Sigma)$, with $\Sigma = (1 - \bar{\alpha}_t)\mathbb{I}$ and $\mu^{\mathrm{obs}} = \mu \odot \mathbf{m}$. It is not hard to see $\nabla_{\mathbf{x}^{\mathrm{obs}}(t)} \log p(\mathbf{x}^{\mathrm{obs}}(t)|\mathbf{x}^{\mathrm{obs}}(0)) \odot \mathbf{m} = -(\mathbf{x}^{\mathrm{obs}}(t) - \mu^{\mathrm{obs}}) \odot \mathbf{m} = -(\mathbf{x}(t) - \mu) \odot \mathbf{m} = \nabla_{\mathbf{x}(t)} \log p(\mathbf{x}(t)|\mathbf{x}(0)) \odot \mathbf{m}$.

(vii) $\exists C > 0 \forall \mathbf{x} \in \mathbb{R}^d, t \in [0, T] : \|\nabla_{\mathbf{x}} \log p_t(\mathbf{x})\|_2 \leq C(1 + \|\mathbf{x}\|_2)$.

(viii) $\exists C > 0, \forall \mathbf{x}, \mathbf{y} \in \mathbb{R}^d : \|\nabla_{\mathbf{x}} \log p_t(\mathbf{x}) - \nabla_{\mathbf{y}} \log p_t(\mathbf{y})\|_2 \leq C\|\mathbf{x} - \mathbf{y}\|_2$.

(ix) $\exists C > 0 \forall \mathbf{x} \in \mathbb{R}^d, t \in [0, T] : \|\mathbf{s}_{\boldsymbol{\theta}}(\mathbf{x}, t)\|_2 \leq C(1 + \|\mathbf{x}\|_2)$.

(x) $\exists C > 0, \forall \mathbf{x}, \mathbf{y} \in \mathbb{R}^d : \|\mathbf{s}_{\boldsymbol{\theta}}(\mathbf{x}, t) - \mathbf{s}_{\boldsymbol{\theta}}(\mathbf{y}, t)\|_2 \leq C\|\mathbf{x} - \mathbf{y}\|_2$.

(xi) Novikov's condition: $\mathbb{E}[\exp(\frac{1}{2} \int_0^T \|\nabla_{\mathbf{x}} \log p_t(\mathbf{x}) - \mathbf{s}_{\boldsymbol{\theta}}(\mathbf{x}, t)\|_2^2 \, \mathrm{d}t)] < \infty$.

(xii) $\forall t \in [0, T], \exists k > 0 : p_t(\mathbf{x}) = O(e^{-\|\mathbf{x}\|_2^k})$ as $\|\mathbf{x}\|_2 \to \infty$.

We mainly follow the proof strategy in Song et al. (2021a). Consider the predefined SDE on the observed data,

$$\mathrm{d}\mathbf{x}^{\mathrm{obs}} = f(\mathbf{x}^{\mathrm{obs}}, t)\mathrm{d}t + g(t)\mathrm{d}\mathbf{w}, \tag{5}$$

and the SDE parametrized by $\theta$,

$$\mathrm{d}\hat{\mathbf{x}}_{\theta}^{\mathrm{obs}} = \mathbf{s}_{\boldsymbol{\theta}}(\hat{\mathbf{x}}_{\theta}^{\mathrm{obs}}, t)\mathrm{d}t + g(t)\mathrm{d}\mathbf{w}. \tag{6}$$

Let $\boldsymbol{\mu}$ and $\boldsymbol{\nu}$ denote the path measure of $\{\mathbf{x}^{\mathrm{obs}}(t)\}_{t \in [0,T]}$ and $\{\hat{\mathbf{x}}_{\theta}^{\mathrm{obs}}(t)\}_{t \in [0,T]}$, respectively. Therefore, the distribution of $p_0(\mathbf{x})$ and $p_{\theta}(\mathbf{x})$ can be represented by the Markov kernel $K(\{\mathbf{z}(t)\}_{t \in [0,T]}, \mathbf{y}) := \delta(\mathbf{z}(0) = \mathbf{y})$ as follow:

$$p_0(\mathbf{x}) = \int K(\{\mathbf{x}^{\mathrm{obs}}(t)\}_{t \in [0,T]}, \mathbf{x})\mathrm{d}\boldsymbol{\mu}(\{\mathbf{x}^{\mathrm{obs}}(t)\}_{t \in [0,T]}),$$

$$p_{\theta}(\mathbf{x}) = \int K(\{\hat{\mathbf{x}}_{\theta}^{\mathrm{obs}}(t)\}_{t \in [0,T]}, \mathbf{x})\mathrm{d}\boldsymbol{\nu}(\{\hat{\mathbf{x}}_{\theta}^{\mathrm{obs}}(t)\}_{t \in [0,T]}).$$

According to the data processing inequality with this Markov kernel, the Kullback–Leibler (KL) divergence between the distribution of $p_0(\mathbf{x})$ and $p_{\theta}(\mathbf{x})$ can be upper bounded, i.e.,

$$D_{\mathrm{KL}}(p_0\|p_{\theta}) = D_{\mathrm{KL}}\left(\int K(\{\mathbf{x}^{\mathrm{obs}}(t)\}_{t \in [0,T]}, \mathbf{x})\mathrm{d}\boldsymbol{\mu} \middle\| \int K(\{\hat{\mathbf{x}}_{\theta}^{\mathrm{obs}}(t)\}_{t \in [0,T]}, \mathbf{x})\mathrm{d}\boldsymbol{\nu}\right) \leq D_{\mathrm{KL}}(\boldsymbol{\mu}\|\boldsymbol{\nu}). \tag{7}$$

By the chain rule of KL divergences,

$$D_{\mathrm{KL}}(\boldsymbol{\mu}\|\boldsymbol{\nu}) = D_{\mathrm{KL}}(p_T\|\pi) + \mathbb{E}_{\mathbf{z} \sim p_T}[D_{\mathrm{KL}}(\boldsymbol{\mu}(\cdot \mid \mathbf{x}^{\mathrm{obs}}(T) = \mathbf{z})\|\boldsymbol{\nu}(\cdot \mid \hat{\mathbf{x}}_{\theta}^{\mathrm{obs}}(T) = \mathbf{z}))]. \tag{8}$$

Under assumptions (i) (iii) (iv) (v) (vi) (vii) (viii), the SDE in Eq (5) has a corresponding reverse-time SDE given by

$$\mathrm{d}\mathbf{x}^{\mathrm{obs}} = [f(\mathbf{x}^{\mathrm{obs}}, t) - g(t)^2 \nabla_{\mathbf{x}^{\mathrm{obs}}} \log p_t(\mathbf{x}^{\mathrm{obs}})]\mathrm{d}t + g(t)\mathrm{d}\overline{\mathbf{w}}. \tag{9}$$

Since Eq (9) is the time reversal of Eq (5), it induces the same path measure $\boldsymbol{\mu}$. As a result, $D_{\mathrm{KL}}(\boldsymbol{\mu}(\cdot \mid \mathbf{x}^{\mathrm{obs}}(T) = \mathbf{z})\|\boldsymbol{\nu}(\cdot \mid \hat{\mathbf{x}}_{\theta}^{\mathrm{obs}}(T) = \mathbf{z}))$ can be viewed as the KL divergence between the path measures induced by the following two (reverse-time) SDEs:

$$\mathrm{d}\mathbf{x}^{\mathrm{obs}} = [f(\mathbf{x}^{\mathrm{obs}}, t) - g(t)^2 \nabla_{\mathbf{x}^{\mathrm{obs}}} \log p_t(\mathbf{x}^{\mathrm{obs}})]\mathrm{d}t + g(t)\mathrm{d}\overline{\mathbf{w}}, \quad \mathbf{x}^{\mathrm{obs}}(T) = \mathbf{x}^{\mathrm{obs}},$$

$$\mathrm{d}\hat{\mathbf{x}}^{\mathrm{obs}} = [f(\hat{\mathbf{x}}^{\mathrm{obs}}, t) - g(t)^2 \mathbf{s}_{\boldsymbol{\theta}}(\hat{\mathbf{x}}^{\mathrm{obs}}, t)]\mathrm{d}t + g(t)\mathrm{d}\overline{\mathbf{w}}, \quad \hat{\mathbf{x}}_{\theta}^{\mathrm{obs}}(T) = \mathbf{x}^{\mathrm{obs}}.$$

Under assumptions (vii) (viii) (ix) (x) (xi), we apply the Girsanov Theorem II [(Øksendal, 1987), Theorem 8.6.6], together with the martingale property of Itô integrals, which yields

$$D_{\mathrm{KL}}(\boldsymbol{\mu}(\cdot \mid \mathbf{x}^{\mathrm{obs}}(T) = \mathbf{z})\|\boldsymbol{\nu}(\cdot \mid \hat{\mathbf{x}}_{\theta}^{\mathrm{obs}}(T) = \mathbf{z}))$$

$$= \mathbb{E}_{\boldsymbol{\mu}}[\frac{1}{2} \int_0^T g(t)^2 \|\nabla_{\mathbf{x}^{\mathrm{obs}}(t)} \log p_t(\mathbf{x}^{\mathrm{obs}}(t)) - \mathbf{s}_{\boldsymbol{\theta}}(\mathbf{x}^{\mathrm{obs}}(t), t)\|_2^2 \, \mathrm{d}t]$$

$$\leq \frac{1}{2(1 - \rho_{\max})} \int_0^T \mathbb{E}_{p_t(\mathbf{x}^{\mathrm{obs}}(t))}[g(t)^2 \|\nabla_{\mathbf{x}^{\mathrm{obs}}(t)} \log p_t(\mathbf{x}^{\mathrm{obs}}(t)) - \mathbf{s}_{\boldsymbol{\theta}}(\mathbf{x}^{\mathrm{obs}}(t), t) \odot \sqrt{\mathbf{1} - \boldsymbol{\rho}}\|_2^2]\mathrm{d}t$$

$$= \frac{1}{2(1 - \rho_{\max})} \int_0^T \mathbb{E}_{p_t(\mathbf{x}^{\mathrm{obs}}(t))}[g(t)^2 \|\nabla_{\mathbf{x}^{\mathrm{obs}}(t)} \log p_t(\mathbf{x}^{\mathrm{obs}}(t)) - \mathbf{s}_{\boldsymbol{\theta}}(\mathbf{x}^{\mathrm{obs}}(t), t) \odot \mathbf{m}\|_2^2]\mathrm{d}t$$

$$= \frac{1}{1 - \rho_{\max}} J_{\mathrm{SM}}(\theta; g(\cdot)^2),$$

$$\tag{10}$$

where $\rho_{\max} = \max_{i=1,\dots,d} \rho_i$ and $1 - \rho_{\max} > 0$ by assumption. Combining Eqs. (7), (8) and (10), we have $D_{\mathrm{KL}}(p_0 \| p_{\boldsymbol{\theta}}) \le \frac{1}{1-\rho_{\max}} J_{\mathrm{SM}}(\boldsymbol{\theta}; g(\cdot)^2) + D_{\mathrm{KL}}(p_T \| \pi)$, which further yields $-\mathbb{E}_{p(\mathbf{x}^{\mathrm{obs}})}[\log p_{\theta}(\mathbf{x})] \le \frac{1}{1-\rho_{\max}} J_{\mathrm{DSM}}(\boldsymbol{\theta}; g(\cdot)^2) + C_1$ by Lemma A.2, where $C_1$ is a constant independent of $\theta$.

**Lemma A.2.** *Denosing Score Matching on missing data is equivalent to Score Matching on missing data, i.e.,*

$$
\begin{aligned}
&\mathbb{E}_{p_t(\mathbf{x}^{obs})}[\|(\mathbf{s}_{\boldsymbol{\theta}}(\mathbf{x}_t^{obs}, t) - \nabla_{\mathbf{x}^{obs}} \log p_t(\mathbf{x}_t^{obs})) \odot \mathbf{m}\|_2^2] \\
&= \mathbb{E}_{p(\mathbf{x}_0^{obs})} \mathbb{E}_{p(\mathbf{x}_t^{obs}|\mathbf{x}_0^{obs})}[\|(\mathbf{s}_{\boldsymbol{\theta}}(\mathbf{x}_t^{obs}, t) - \nabla_{\mathbf{x}^{obs}} \log p(\mathbf{x}_t^{obs} \mid \mathbf{x}_0^{obs})) \odot \mathbf{m}\|_2^2] + C,
\end{aligned}
\tag{11}
$$

*where* $\mathbf{m} = \mathbb{1}\{\mathbf{x}_0^{obs} \ne \mathrm{na}\}$ *indicated the missing entries in* $\mathbf{x}^{obs}$ *and* $C$ *is a constant that does not depend on* $\boldsymbol{\theta}$. *We interchange* $\mathbf{x}^{obs}(t)$ *with* $\mathbf{x}_t^{obs}$.

*Proof.* We begin with the Score Matching on the left-hand side of (11)

$$
\begin{aligned}
\mathrm{LHS} &= \mathbb{E}_{p_t(\mathbf{x}_t^{\mathrm{obs}})}[\|(\mathbf{s}_{\boldsymbol{\theta}}(\mathbf{x}_t^{\mathrm{obs}}, t) - \nabla_{\mathbf{x}_t^{\mathrm{obs}}} \log p_t(\mathbf{x}_t^{\mathrm{obs}})) \odot \mathbf{m}\|_2^2] \\
&= \mathbb{E}_{p_t(\mathbf{x}_t^{\mathrm{obs}})}[\|\mathbf{s}_{\boldsymbol{\theta}}(\mathbf{x}_t^{\mathrm{obs}}, t) \odot \mathbf{m}\|^2] - S(\theta) + C_2,
\end{aligned}
\tag{12}
$$

where $C_2 = \mathbb{E}_{p_t(\mathbf{x}_t^{\mathrm{obs}})}[\|\nabla_{\mathbf{x}_t^{\mathrm{obs}}} \log p_t(\mathbf{x}_t^{\mathrm{obs}}) \odot \mathbf{m}\|^2]$ is a constant that does not depend on $\boldsymbol{\theta}$, and

$$
\begin{aligned}
S(\theta) &= 2\mathbb{E}_{p_t(\mathbf{x}_t^{\mathrm{obs}})}[\langle \mathbf{s}_{\boldsymbol{\theta}}(\mathbf{x}_t^{\mathrm{obs}}, t), \nabla_{\mathbf{x}_t^{\mathrm{obs}}} \log p_t(\mathbf{x}_t^{\mathrm{obs}}) \odot \mathbf{m}\rangle] \\
&= 2\int_{\mathbf{x}_t^{\mathrm{obs}}} p_t(\mathbf{x}_t^{\mathrm{obs}}) \langle \mathbf{s}_{\boldsymbol{\theta}}(\mathbf{x}_t^{\mathrm{obs}}, t), \nabla_{\mathbf{x}_t^{\mathrm{obs}}} \log p_t(\mathbf{x}_t^{\mathrm{obs}}) \odot \mathbf{m}\rangle \, \mathrm{d}\mathbf{x}_t^{\mathrm{obs}} \\
&= 2\int_{\mathbf{x}_t^{\mathrm{obs}}} \langle \mathbf{s}_{\boldsymbol{\theta}}(\mathbf{x}_t^{\mathrm{obs}}, t), \nabla_{\mathbf{x}_t^{\mathrm{obs}}} p_t(\mathbf{x}_t^{\mathrm{obs}}) \odot \mathbf{m}\rangle \, \mathrm{d}\mathbf{x}_t^{\mathrm{obs}} \\
&= 2\int_{\mathbf{x}_t^{\mathrm{obs}}} \langle \mathbf{s}_{\boldsymbol{\theta}}(\mathbf{x}_t^{\mathrm{obs}}, t), \frac{\mathrm{d}}{\mathrm{d}\mathbf{x}_t^{\mathrm{obs}}} \int_{\mathbf{x}_0^{\mathrm{obs}}} p_0(\mathbf{x}_0^{\mathrm{obs}}) p(\mathbf{x}_t^{\mathrm{obs}} \mid \mathbf{x}_0^{\mathrm{obs}}) \odot \mathbf{m} \, \mathrm{d}\mathbf{x}_0^{\mathrm{obs}}\rangle \, \mathrm{d}\mathbf{x}_t^{\mathrm{obs}} \\
&= 2\int_{\mathbf{x}_t^{\mathrm{obs}}} \int_{\mathbf{x}_0^{\mathrm{obs}}} p_0(\mathbf{x}_0^{\mathrm{obs}}) p(\mathbf{x}_t^{\mathrm{obs}} \mid \mathbf{x}_0^{\mathrm{obs}}) \langle \mathbf{s}_{\boldsymbol{\theta}}(\mathbf{x}_t^{\mathrm{obs}}, t), \frac{\mathrm{d} \log p(\mathbf{x}_t^{\mathrm{obs}} \mid \mathbf{x}_0^{\mathrm{obs}})}{\mathrm{d}\mathbf{x}_t^{\mathrm{obs}}} \odot \mathbf{m}\rangle \, \mathrm{d}\mathbf{x}_0^{\mathrm{obs}} \mathrm{d}\mathbf{x}_t^{\mathrm{obs}} \\
&= 2\mathbb{E}_{p(\mathbf{x}_t^{\mathrm{obs}}, \mathbf{x}_0^{\mathrm{obs}})}[\langle \mathbf{s}_{\boldsymbol{\theta}}(\mathbf{x}_t^{\mathrm{obs}}, t), \frac{\mathrm{d} \log p(\mathbf{x}_t^{\mathrm{obs}} \mid \mathbf{x}_0^{\mathrm{obs}})}{\mathrm{d}\mathbf{x}_t^{\mathrm{obs}}} \odot \mathbf{m}\rangle].
\end{aligned}
$$

Substituting this expression for $S(\theta)$ into Eq (12) yields

$$
\begin{aligned}
\mathrm{LHS} &= \mathbb{E}_{p_t(\mathbf{x}_t^{\mathrm{obs}})}[\|\mathbf{s}_{\boldsymbol{\theta}}(\mathbf{x}_t^{\mathrm{obs}}, t) \odot \mathbf{m}\|^2] \\
&\quad - 2\mathbb{E}_{p(\mathbf{x}_t^{\mathrm{obs}}, \mathbf{x}_0^{\mathrm{obs}})}[\langle \mathbf{s}_{\boldsymbol{\theta}}(\mathbf{x}_t^{\mathrm{obs}}, t), \frac{\mathrm{d} \log p(\mathbf{x}_t^{\mathrm{obs}} \mid \mathbf{x}_0^{\mathrm{obs}})}{\mathrm{d}\mathbf{x}_t^{\mathrm{obs}}} \odot \mathbf{m}\rangle] + C_2.
\end{aligned}
\tag{13}
$$

On the other hand, we also have the Denoising Score Matching objective on the right-hand side of (11) is

$$
\begin{aligned}
\mathrm{RHS} &= \mathbb{E}_{p_t(\mathbf{x}_t^{\mathrm{obs}})}[\|\mathbf{s}_{\boldsymbol{\theta}}(\mathbf{x}_t^{\mathrm{obs}}, t) \odot \mathbf{m}\|^2] \\
&\quad - 2\mathbb{E}_{p(\mathbf{x}_t^{\mathrm{obs}}, \mathbf{x}_0^{\mathrm{obs}})}[\langle \mathbf{s}_{\boldsymbol{\theta}}(\mathbf{x}_t^{\mathrm{obs}}, t), \frac{\mathrm{d} \log p_t(\mathbf{x}_t^{\mathrm{obs}} \mid \mathbf{x}_0^{\mathrm{obs}})}{\mathrm{d}\mathbf{x}_t^{\mathrm{obs}}}\rangle \odot \mathbf{m}] + C_3,
\end{aligned}
\tag{14}
$$

where $C_3 = \mathbb{E}_{p(\mathbf{x}_t^{\mathrm{obs}}, \mathbf{x}_0^{\mathrm{obs}})}[\|\frac{\mathrm{d} \log p_t(\mathbf{x}_t^{\mathrm{obs}}|\mathbf{x}_0^{\mathrm{obs}})}{\mathrm{d}\mathbf{x}_t^{\mathrm{obs}}} \odot \mathbf{m}\|^2] + C$ is a constant that does not depend on $\boldsymbol{\theta}$.

Comparing equations (13) and (14), we thus show that the two optimization objectives are equivalent up to a constant. $\qquad\square$

# B MORE DETAILS ON EXPERIMENTS

## B.1 DATASETS FOR IMPUTATION TASK

The detailed description of the dataset can be found in Table 5, which specifies the number of training data (#Train), the number of testing data (#Test), the number of categorical (discrete) variables in the tabular dataset (#Categorical), and the number of continuous variables (#Continuous).

Table 5: Real-World Datasets Used in Imputation Tasks.

| Dataset | #Train | #Test | #Categorical | #Continuous |
|---|---|---|---|---|
| Census Income Dataset (Census) (Kohavi & Becker, 1996) | 16000 | 4000 | 9 | 6 |
| Breast Cancer Wisconsin (Breast) (WIlliam, 1992) | 560 | 139 | 0 | 10 |
| Wine Quality (Wine) (Paulo et al., 2009) | 3918 | 980 | 0 | 12 |
| Concrete Compressive Strength (Concrete) (I-Cheng, 2007) | 824 | 206 | 0 | 9 |
| Libras Movement (Libras) (Daniel et al., 2009) | 288 | 72 | 0 | 91 |
| Diabetes Dataset (Kohavi & Becker) | 16000 | 4000 | 15 | 7 |

**Missing mechanism**  We adopt the same missing mechanism as Zheng & Charoenphakdee (2022), i.e., MCAR with missing ratio as 0.2. To be more precisely, the detailed implementation of MCAR is the "Row Missing" defined in paragraph B.2.

## B.2  DATASETS FOR GENERATION TASK

The detailed description of the dataset can be found in Table 6, which specifies the number of training data (#Train), the number of testing data (#Test), the number of categorical (discrete) variables in the tabular dataset (#Categorical), and the number of continuous variables (#Continuous). Moreover, the last column shows the evaluation task we adopted as detailed later. The details of the data generated from a Bayesian Network can be found in Appendix B.2.

Table 6: Synethetic and Real-World Datasets Used in Generation Tasks.

| Dataset | #Train | #Test | #Categorical | #Continuous | Utility |
|---|---|---|---|---|---|
| Bayesian Network | 20000 | 2000 | 3 | 2 | Multi-class classification |
| Census (Kohavi & Becker, 1996) | 16000 | 4000 | 9 | 6 | Binary classification |
| MIMIC4ED (Xie et al., 2022) | 353150 | 88287 | 46 | 27 | Regression |

**Details of the Bayesian Network**  Figure 2 demonstrates the Bayesian Network for generating the tabular data. It contains two continuous variables C1, C2, and three discrete random variables D1, D2, and D3. The distribution of these variables is set as follows. The marginal distribution of C1 is $\mathcal{N}(25, 2)$, the conditional distribition of C2 given C1 is $C2|C1 \sim \mathcal{N}(0.1 \cdot C1 + 50, 5)$, and the marginal distribution of D1 is $Bernoulli(0.3)$, where $Bernoulli(\xi)$ stands for the Bernoulli distribution with mean equal to $\xi$. The conditional distribution of D2, given C1, C2 and D1, is set as

$$D2|C1, C2, D1 \sim \begin{cases} Ca(0.3, 0.6, 0.1) & C1 > 26, C2 > 55, D1 = 1; \\ Ca(0.2, 0.3, 0.5) & C1 > 26, C2 \leq 55, D1 = 1; \\ Ca(0.7, 0.1, 0.2) & C1 \leq 26, C2 > 55, D1 = 1; \\ Ca(0.1, 0.2, 0.7) & C1 \leq 26, C2 \leq 55, D1 = 1; \\ Ca(0.05, 0.05, 0.9) & D1 = 0, \end{cases}$$

where $Ca(p1, p2, 1 - p1 - p2)$ denotes the categorical (discrete) distribution for three pre-specified categories. The conditional distribution of D3 given D2 is

$$D3|D2 \sim \begin{cases} Bernoulli(0.2) & D2 = 0; \\ Bernoulli(0.4) & D2 = 1; \\ Bernoulli(0.8) & D2 = 2. \end{cases}$$

**Choice of Masks under Different Missing Mechanisms**  To evaluate the performance of *MissDiff* on different missing mechanisms, we give a detailed explanation of the practical implementation of MCAR (Li et al., 2019; Yoon et al., 2018a), MAR(Ipsen et al., 2022; Li & Marlin, 2020), and NMAR (Muzellec et al., 2020; Ipsen et al., 2021).

- MCAR: there are three types of missing mechanisms in MCAR.

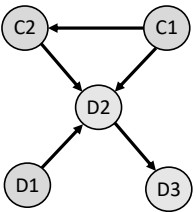

Figure 2: The demonstration of the Bayesian Network for generating the tabular data. "C1" and "C2" denote the continuous variables and "D1", "D2", "D3" denotes the discrete random variables. The marginal/conditional distributions for each node are detailed in Section B.2.

- Row Missing. For a given missing ratio $\alpha \in (0, 1)$, we have the number of elements missing in each row (i.e., for each sample $\mathbf{x}_i$) is $\lfloor d\alpha \rfloor$, where $\lfloor z \rfloor$ is the greatest integer less than $z$, and the location/index of the missing entries is randomly chosen according to the uniform distribution.

- Column Missing. For a given missing ratio $\alpha$, we have the number of elements missing in each column (for each feature) is $\lfloor n\alpha \rfloor$, and the location/index of the missing entries is randomly chosen according to the uniform distribution.

- Independent Missing. Each entry in the table is masked missing according to the realization of a Bernoulli random variable with parameter $\alpha$.

- MAR: a fixed subset of variables that cannot have missing values is first sampled. Then, the remaining variables will have missing values according to a logistic model with random weights, which takes the non-missing variables as inputs. The outcome of this logistic model is re-scaled to attain a given missing ratio $\alpha$.

- NMAR: the same pipeline as MAR with the inputs of the logistic model are masked by the MCAR mechanism. We refer to Muzellec et al. (2020) for more detailed explanations.

*Remark* B.1. Under the three missing mechanisms in MCAR, with the missing ratio parameter set as $0 < \alpha < 1$, the condition in Theorem 3.2 can be satisfied with probability at least $1 - \delta$, where $\delta = \max\{(\frac{\alpha d - 1}{d})^n d, \alpha, \alpha^n d\}$ and it will be sufficiently small when $\alpha$ is small and $n$ is sufficiently large.

Remark B.1 gives the guarantee that *MissDiff* can recover the oracle score under MCAR with high probability. In Sections 4.3, we adopt the missing ratio $\alpha = 0.2$ and XGBoost for the downstream tasks as the default setting. More experimental results can be found in Appendix B.5.

### B.3 ALGORITHMS FOR IMPUTATION AND GENERATION TASKS

**Algorithm for Generating** *MissDiff* adopts the algorithm 2 for imputation task and algorithm 3 for generating new complete data. For the imputation, the noising version of the observed data is used as the guidance. This element-wise multiplication guarantees the output $\mathbf{x}_0$ has the same value as $\mathbf{x}_{\text{obs}}$ in the observed entries.

---

**Algorithm 2** *MissDiff* for Imputation

---

**Require:** Observed data $\mathbf{x}_0^{\text{obs}}$, Diffusion model $\mathbf{s}_{\boldsymbol{\theta}}$, hyperparameter $\beta_t, \sigma_t$, denote $\alpha_t = 1 - \beta_t$ and $\bar{\alpha}_t = \prod_{s=1}^{t} \alpha_s$.
1: Sample $\mathbf{x}_T \sim \mathcal{N}(\mathbf{0}, \mathbb{I})$;
2: Infer mask $\mathbf{m} = \mathbb{1}[\mathbf{x}_0^{\text{obs}} \neq \text{na}]$;
3: $t = T$;
4: **while** $t \neq 0$ **do**
5:     Sample $\epsilon_t^{\text{obs}} \sim \mathcal{N}(\mathbf{0}, \mathbb{I})$ if $t > 1$, else $\epsilon_t^{\text{obs}} = \mathbf{0}$;
6:     $\mathbf{x}_{t-1}^{\text{obs}} = \sqrt{\bar{\alpha}_{t-1}} \mathbf{x}_0^{\text{obs}} + (1 - \bar{\alpha}_{t-1}) \epsilon_t^{\text{obs}}$
7:     Sample $\epsilon_t \sim \mathcal{N}(\mathbf{0}, \mathbb{I})$ if $t > 1$, else $\epsilon_t = \mathbf{0}$;
8:     $\mathbf{x}_{t-1} = \frac{1}{\sqrt{\alpha_t}}(\mathbf{x}_t - \frac{\beta_t}{\sqrt{1-\bar{\alpha}_t}} \mathbf{s}_{\boldsymbol{\theta}}(\mathbf{x}_t, t)) + \sigma_t \epsilon_t$;
9:     $\mathbf{x}_{t-1} = \mathbf{m} \odot \mathbf{x}_{t-1}^{\text{obs}} + (\mathbf{1} - \mathbf{m}) \odot \mathbf{x}_{t-1}$
10:    $t = t - 1$;
11: **end while**
12: **return** $\mathbf{x}_0$.

---

**Algorithm 3** *MissDiff* for Generation

---

**Require:** Diffusion model $\mathbf{s}_{\boldsymbol{\theta}}$, hyperparameter $\beta_t, \sigma_t$, denote $\alpha_t = 1 - \beta_t$ and $\bar{\alpha}_t = \prod_{s=1}^{t} \alpha_s$.
1: Sample $\mathbf{x}_T \sim \mathcal{N}(\mathbf{0}, \mathbb{I})$;
2: $t = T$;
3: **while** $t \neq 0$ **do**
4:     Sample $\epsilon_t \sim \mathcal{N}(\mathbf{0}, \mathbb{I})$ if $t > 1$, else $\epsilon_t = \mathbf{0}$;
5:     $\mathbf{x}_{t-1} = \frac{1}{\sqrt{\alpha_t}}(\mathbf{x}_t - \frac{\beta_t}{\sqrt{1-\bar{\alpha}_t}} \mathbf{s}_{\boldsymbol{\theta}}(\mathbf{x}_t, t)) + \sigma_t \epsilon_t$;
6:     $t = t - 1$;
7: **end while**
8: **return** $\mathbf{x}_0$.

---

### B.4 IMPLEMENTATION DETAILS

To make the transition $p(\mathbf{x}^{\text{obs}}(t)|\mathbf{x}^{\text{obs}}(0))$ and the gradient $\nabla_{\mathbf{x}^{\text{obs}}(t)} \log p(\mathbf{x}^{\text{obs}}(t) \mid \mathbf{x}^{\text{obs}}(0))$ well defined for the mixed-type data, we use 0 to replace na for continuous variables and a new category to represent na for discrete variables, which is the same operation as in Nazábal et al. (2018); Ma et al. (2020) that can help to feed fixed dimensional data into neural networks. One-hot embedding is applied to discrete variables.

We adopt four layers residual network as the backbone of the diffusion model. The dimension of the diffusion embedding is 128 with channels as 64. We set the minimum noise level $\beta_1 = 0.0001$ and the maximum noise level $\beta_T = 0.5$ in Algorithm 1 and Algorithm 3 with quadratic schedule

$$\beta_t = \left( \frac{T-t}{T-1} \sqrt{\beta_1} + \frac{t-1}{T-1} \sqrt{\beta_T} \right)^2.$$

We mainly follow the hyperparameter in the previous works that train the diffusion model on tabular data Tashiro et al. (2021); Zheng & Charoenphakdee (2022). We use the Adam optimizer with MultiStepLR with 0.1 decay at $25\%, 50\%, 75\%$, and $90\%$ of the total epochs and with an initial learning rate as 0.0005.

With regard to the baselines methods in Table 1, we either adopt the results and hyperparameters from Zheng & Charoenphakdee (2022) or use the open source implementation from `https://github.com/vanderschaarlab/hyperimpute` for choosing the corresponding hyperparameters.

With regard to the baselines of STaSy, we adopt the same setting of its open resource implementation [8], i.e., Variance Exploding SDE with six layers ConcatSquash network as the backbone of the diffusion model and Fourier embedding, the adam optimizer with learning rate as 2e-03, training with batch size 64 and 250 epochs/1000 epochs with additional 50 finetuning epochs.

---

[8] `https://openreview.net/forum?id=1mNssCWt_v`

For the downstream classifier/regressor, we adopt the same base hyperparameters in [Kim et al. (2023), Table 26].

## B.5 Additional Experiential Results

### B.5.1 Additional Results for Fidelity Evaluation

Table 7, 8, and 9 provide SDMetrics metric evaluation on *MissDiff*. They correspond to Table 2, 3, and 4 in Section 4.3.2.

Table 7: *Fidelity* evaluation of *MissDiff* on Census dataset. The *larger* the score, the *better* the overall quality of synthetic data is.

| | *MissDiff* | *Diff-delete* | *Diff-mean* | *STaSy-delete* | *STaSy-mean* | CSDI_T |
|---|---|---|---|---|---|---|
| Row Missing | **80.59**% | - | 76.92% | - | 56.75% | 77.60% |
| Column Missing | **82.70**% | 75.03% | 76.17% | 56.90% | 51.54% | 73.84% |
| Independent Missing | **83.16**% | 74.94% | 76.60% | 56.07% | 57.06% | 82.56% |

Table 8: *Fidelity* evaluation of *MissDiff* on MIMIC4ED dataset. *Diff-delete* and *STaSy-delete* cannot be applied since no data $\mathbf{x}_i$ will be left after deleting the incomplete data.

| | *MissDiff* | *Diff-mean* | *STaSy-mean* |
|---|---|---|---|
| Row Missing | **84.45**% | 75.22% | 82.94% |
| Column Missing | **79.24**% | 76.57% | 79.03% |
| Independent Missing | **78.01**% | 76.16% | 77.21% |

Table 9: *Fidelity* evaluation of *MissDiff* on Census dataset under MAR, NMAR.

| | *MissDiff* | Diff-delete | Diff-mean | *STaSy-delete* | *STaSy-mean* | CSDI_T |
|---|---|---|---|---|---|---|
| MAR | 77.45% | 73.78% | 76.08% | 57.51% | 50.06% | **78.14**% |
| NMAR | **77.88**% | 75.72% | 76.97% | 54.11% | 50.6% | 77.51% |

### B.5.2 Additional Results of Other Criteria for *Utility* Evaluation

Table 10, 11, and 12 provide the additional experimental results for other criteria under *Utility* evaluation for Table 2, 3, and 4 in the main paper, i.e., the F1, Weighted-F1, AUROC for the classification task and $R^2$ for the regression task. A detailed explanation of the above-mentioned criteria can be found in Kim et al. (2023). To make our paper self-contained, we briefly restate it here.

1. Binary F1 for binary classification: sklearn.metrics.f1_score with 'average'='binary'.
2. Macro F1 for multi-class classification: sklearn.metrics.f1_score with 'average'='macro'.
3. Weighted-F1: $= \sum_{i=0}^{K} w_i s_i$, where $K$ denotes the number of classes, the weight of $i$-th class $w_i$ is $\frac{1-p_i}{K-1}$, $p_i$ is the proportion of $i$-th class's cardinality in the whole dataset, and score $s_i$ is a per-class F1 of $i$-th class (in a One-vs-Rest manner).
4. AUROC: sklearn.metrics.roc_auc_score.

From the results in Table 10, 11, and 12, it can be seen that the proposed *MissDiff* consistently outperforms the compared methods in most instances. For the column missing case, *MissDiff* tends to perform worse, which indicates the potential limitations of the proposed method for future investigations.

### B.5.3 Experiment Results for Different Classifiers/Regressors

As mentioned in section 4.2, we train various models, including Decision Tree, AdaBoost, Logistic/Linear Regression, MLP classifier/regressor, RandomForest, and XGBoost, on synthetic data.

Table 10: *Utility* evaluation of *MissDiff* on Census dataset with other criteria.

| Criterion | Missing Mechanism | *MissDiff* | *Diff-delete* | *Diff-mean* | *STaSy-delete* | *STaSy-mean* |
|---|---|---|---|---|---|---|
| Binary F1 | Row Missing | **0.344** | - | 0.280 | - | 0.314 |
| | Column Missing | 0.141 | 0.063 | 0.413 | **0.509** | 0.383 |
| | Independent Missing | **0.291** | 0.045 | 0.225 | 0.274 | 0.241 |
| Weighted-F1 | Row Missing | 0.470 | - | 0.423 | - | **0.488** |
| | Column Missing | 0.305 | 0.249 | 0.523 | **0.571** | 0.490 |
| | Independent Missing | **0.431** | 0.237 | 0.375 | 0.416 | 0.389 |
| AUROC | Row Missing | **0.772** | - | 0.685 | - | 0.731 |
| | Column Missing | 0.539 | 0.469 | **0.757** | 0.750 | 0.637 |
| | Independent Missing | **0.650** | 0.474 | 0.655 | 0.621 | 0.613 |

Table 11: *Utility* evaluation of *MissDiff* on MIMIC4ED dataset with $R^2$ criterion. *Diff-delete* and *STaSy-delete* cannot be applied since no data $\mathbf{x}_i$ will be left after deleting the incomplete data.

| Missing mechanism | *MissDiff* | *Diff-mean* | *STaSy-mean* |
|---|---|---|---|
| Row Missing | **0.088** | 0.057 | 0.067 |
| Column Missing | **0.095** | 0.023 | 0.073 |
| Independent Missing | **0.156** | 0.062 | 0.142 |

Table 12: *Utility* evaluation of *MissDiff* on Census dataset under MAR, NMAR with other criteria.

| Criterion | Missing Mechanism | *MissDiff* | *Diff-delete* | *Diff-mean* |
|---|---|---|---|---|
| Binary F1 | MAR | 0.346 | 0.108 | 0.224 |
| | NMAR | **0.464** | 0.233 | 0.383 |
| Weighted-F1 | MAR | **0.473** | 0.276 | 0.376 |
| | NMAR | **0.564** | 0.364 | 0.501 |
| AUROC | MAR | **0.833** | 0.441 | 0.774 |
| | NMAR | **0.834** | 0.499 | 0.746 |

Table 13 to 17 present the corresponding results on different classifiers/regressors, from which we can see that *MissDiff* still performs well under most cases.

Table 13: *Utility* evaluation of *MissDiff* on Census dataset by Decision Tree.

| | *MissDiff* | *Diff-delete* | *Diff-mean* | *STaSy-delete* | *STaSy-mean* |
|---|---|---|---|---|---|
| Row Missing | **78.08**% | - | 74.55% | - | 60.74% |
| Column Missing | 62.65% | 69.10% | **78.88**% | 65.38% | 66.31% |
| independent | **80.68**% | 72.68% | 67.70% | 76.35% | 55.99% |

Table 14: *Utility* evaluation of *MissDiff* on Census dataset by AdaBoost.

| | *MissDiff* | *Diff-delete* | *Diff-mean* | *STaSy-delete* | *STaSy-mean* |
|---|---|---|---|---|---|
| Row Missing | **80.38**% | - | 79.28% | - | 73.23% |
| Column Missing | 72.18% | 76.30% | **80.65**% | 69.60% | 42.24% |
| independent | **78.70**% | 76.13% | 75.96% | 76.55% | 78.39% |

Table 15: *Utility* evaluation of *MissDiff* on Census dataset by Logistic Regression.

|  | *MissDiff* | *Diff-delete* | *Diff-mean* | *STaSy-delete* | *STaSy-mean* |
|---|---|---|---|---|---|
| Row Missing | **79.20**% | - | 77.08% | - | 71.04% |
| Column Missing | 73.50% | 76.30% | **77.45**% | 66.91% | 69.08% |
| independent | 76.20% | **76.30**% | 76.25% | 77.13% | 69.68% |

Table 16: *Utility* evaluation of *MissDiff* on Census dataset by Multi-layer Perceptron (MLP).

|  | *MissDiff* | *Diff-delete* | *Diff-mean* | *STaSy-delete* | *STaSy-mean* |
|---|---|---|---|---|---|
| Row Missing | **77.70**% | - | 75.13% | - | 49.78% |
| Column Missing | 68.33% | 65.75% | **75.00**% | 70.97% | 58.83% |
| independent | **75.33**% | 72.18% | 74.30% | 76.81% | 37.59% |

Table 17: *Utility* evaluation of *MissDiff* on Census dataset by Random Forest.

|  | *MissDiff* | *Diff-delete* | *Diff-mean* | *STaSy-delete* | *STaSy-mean* |
|---|---|---|---|---|---|
| Row Missing | **80.10**% | - | 77.13% | - | 72.68% |
| Column Missing | 73.68% | 76.33% | **79.88**% | 74.70% | 71.58% |
| independent | **79.33**% | 76.30% | 76.38% | 76.31% | 76.98% |

### B.5.4 ADDITIONAL RESULTS FOR *STaSy-delete* AND *STaSy-mean*

The experimental results of *STaSy-delete* and *STaSy-mean* in Tables 2 and 7 are obtained by training diffusion model for 1000 epochs, compared with 250 epochs of *MissDiff*, *Diff-delete*, and *Diff-mean*. If we train *STaSy-delete* and *STaSy-mean* as the same training epochs (250 epochs) on the Census dataset under MCAR as *MissDiff*, their performance is demonstrated in Table 18 and 19. This observation highlights that the proposed *MissDiff* requires considerably fewer training epochs compared to STaSy in order to achieve satisfactory results when handling data with missing values.

Table 18: *Fidelity* evaluation of *MissDiff* on Census dataset with 250 training epochs.

|  | *MissDiff* | *Diff-delete* | *Diff-mean* | *STaSy-delete* | *STaSy-mean* |
|---|---|---|---|---|---|
| Row Missing | **80.59**% | - | 76.92% | - | 50.08% |
| Column Missing | **82.70**% | 75.03% | 76.17% | 52.49% | 49.63% |
| independent | **83.16**% | 74.94% | 76.60% | 53.7% | 50.11% |

Table 19: *Utility* evaluation of *MissDiff* on Census dataset with 250 training epochs.

|  | *MissDiff* | *Diff-delete* | *Diff-mean* | *STaSy-delete* | *STaSy-mean* |
|---|---|---|---|---|---|
| Row Missing | **79.48**% | - | 78.45% | - | 60.96% |
| Column Missing | 71.68% | 72.89% | **79.60**% | 56.19% | 61.46% |
| independent | **79.49**% | 75.39% | 75.96% | 49.78% | 70.68% |

### B.6 COMPUTATIONAL TIME

All the experiments are conducted on NVIDIA A100 Tensor Core GPUs. It takes around 30 minutes for each experiment on the Bayesian Network, around 5 hours for each experiment on the UCI dataset, and around one day for each experiment on the MIMIC4ED dataset.

