# OpenReview forum: "MissDiff: Training Diffusion Models on Tabular Data with Missing Values"
_ICLR.cc/2024/Conference — Submitted to ICLR 2024_

### Official Review · Reviewer_JZFN · 2023-10-31

**Soundness:** 3 good
**Presentation:** 2 fair
**Contribution:** 3 good
**Rating:** 6
**Confidence:** 3

**Summary:**

In the paper, the authors first comment on the limitations of two existing frameworks: (1) impute and then generate framework is constructing a complete training dataset first, but is biased. (2)generate then impute framework needs MCAR and MAR conditions.  The authors propose a simple MissDiff algorithm that utilizes a masking function on the loss function to handle the missing data values. The authors examine the effectiveness of MissDiff, by theoretically verifying that MissDiff can learn the oracle score on the complete data. Furthermore, MissDiff is an upper bound for the negative likelihood on the observed data.  The experiment shows that MissDiff performs comparably better to various imputation approaches in the imputation tasks, and MissDiff is performing better than impute-and -then-generate or generate-then-impute frameworks training with vanilla diffusion.

**Strengths:**

S1. The methodology is given in a clear manner; the proper notations are used and the authors provide solid explanations, examples and proofs to each of the claims.

S2. The authors have provided a theoretical proof to justify the effectiveness of the masking mechanism by examining the loss functions and verifying that optimal solution of MissDiff can learn the oracle score on the complete data and MissDiff is also the upper bound of the negative likelihood of the generative model on the observed data.

S3. The simulation results seem promising. The authors have provided convincing figures to show that under fidelity evaluation, MissDiff performs better than Diff-delete and Diff-mean under different data missing scenarios.

**Weaknesses:**

W1. The experimental setting for Experiment 1 should be detailed described in the main text or appendix. It is confusing to the reviewer how MissDiff’s RMSE and error rate are provided (if the MissDiff is doing unconditional generation according to Algorithm 2), and what is the training hyper parameters that are associated with MissDiff in this scenario? It is also confusing how RMSE evaluation of MIssDiff on MINIC4ED dataset is acquired (see Table 3). The authors should consider providing more experimental details.

W2. Table 2 shows that MissDiff’s utility is significantly lower than Diff-mean for column missing scenario, the authors should have provided an explanation for the inferior performance of MissDiff for column missing Census data.

W3. In terms of writing, Algorithm 2 is not different from the existing variance preserving sampling method. Therefore, it is redundant to include Algorithm 2 in the main text. The reviewer suggest to move the Algorithm 2 into the appendix.

**Questions:**

Q1. What do the numbers mean in the parenthesis for Table 1?

Q2. The authors mention that they hope to evaluate MissDiff with more complicated data types such as video or language data, but the reviewer is wondering how missing values are defined in videos and images. It seems trivial to handle the missing values in video and images due to the amount of redundant information the videos and images contain. 70% of missing pixels can still generate reliable images in masked auto-encoders. For languages, missing values are also beneficial to generative models. Masked-out tokens are important for language model training,

Q3. The reviewer is wondering how MissDiff can handle the categorical features and nonnumerical features in Tabular Data. While the categorical features and non-numerical features could be treated as continuous features according to Equation 4, the performance is not as good as directly performing discrete diffusion models (D3PM [1], tauLDR[2]). Could MissDiff be adapted to discrete diffusion models?

[1] Austin, Jacob, Daniel D. Johnson, Jonathan Ho, Daniel Tarlow, and Rianne van den Berg. 2023. "Structured Denoising Diffusion Models in Discrete State-Spaces." arXiv preprint.  https://arxiv.org/abs/2107.03006.

[2] Campbell, Andrew, Joe Benton, Valentin De Bortoli, Tom Rainforth, George Deligiannidis, and Arnaud Doucet. 2022. "A Continuous Time Framework for Discrete Denoising Models." arXiv preprint.  https://arxiv.org/abs/2205.14987.

**Details Of Ethics Concerns:**

N/a.

---

> ### Author Response · Authors · 2023-11-20
> **Rebuttal by Authors (1/2)**
>
> We thank the reviewer for the comments, and we appreciate the time you spent on the paper. Below we address the concerns and comments that you have provided.
>
> **Q**: *The experimental setting for Experiment 1 should be detailed described in the main text or appendix.*
>
> **A**: Thank you for your suggestion. The experimental setting for Section 4.1 can be found in Appendix B.1 in the revised paper (updated on openrview).
>
> **Q**: *It is confusing to the reviewer how MissDiff’s RMSE and error rate are provided ... what is the training hyper parameters that are associated with MissDiff in this scenario?*
>
> **A**: Thank you very much for your question. We have added the imputation algorithm of MissDiff in Algorithm 2, which can be found in Appendix B.3. Since MissDiff models the score for complete data distribution, we can adopt the same model and training hyperparameters for imputation and generation tasks. The hyperparameters are described in Appendix B.4.
>
> **Q**: *It is also confusing how RMSE evaluation of MIssDiff on MINIC4ED dataset is acquired (see Table 3).*
>
> **A**: The RMSE in Table 3 is different from imputation tasks. As mentioned in "Evaluation Criterion" paragraph in section 4.2, we follow the evaluation criterion as Xu et al. (2019); Kim et al. (2023); Kotelnikov et al. (2022) that use {\it utility} to evaluate the performance of the generated complete data. We train a downstream model (e.g., XGBoost model) on generated data for regression tasks and report the RMSE of the predicted value against the oracle value on the real (test) data.
>
> **Q**: *Table 2 shows that MissDiff’s utility is significantly lower than Diff-mean for column missing scenario.*
>
> **A**: We believe the column missing mechanism described in Appendix B.1 is actually a special scenario. Most specifically, the mask $\mathbf m$ (indicator of missing values) for each row (sample) would depend on the masks of other rows as well, since the missing rate for each column is fixed. It leads to dependence between missing samples. We further note that in our population objective function eq (4), as a standard practice, we regard the sample pair (m,x) are iid and the expectation in (4) is taken with respect to this joint distribution. When the sample size of the dataset is relatively small, such sample dependence is more evident, and MissDiff is not as good as Diff-mean. However, when there is a sufficiently large number of samples, as in Table 3,8, and 11 for the MIMIC4ED dataset (22 times larger than the Census dataset), such dependence becomes very weak and the joint sample pairs (m,x) are close to independent; in these cases, MissDiff performs better.
>
> **Q**: *It is redundant to include Algorithm 2 in the main text.*
>
> **A**: Thank you for your suggestion. We have moved Algorithm 2 to Appendix B.3. For completeness, we have also added the algorithm for imputation in Appendix B.3.
>
> **Q**: *What do the numbers mean in the parenthesis for Table 1?*
>
> **A**: The numbers are the standard deviation of five independent trials. We add the explanation to the caption of Table 1.
>
> **Q**: *How missing values are defined in videos and images? How to evaluate MissDiff with more complicated data types such as video or language data?*
>
> **A**: Thank you very much for your question.
>
> - For video and image data, we can either regard the missing frames or the missing pixels in each frame as the missing data. It is a good point of view that the amount of redundant information in video and images is high. We think MissDiff would still be useful against VAE-based or GAN-based methods since our approach can learn the complete data distribution from incomplete data (low-quality video data) and generate complete data in a unified framework.
> - For the language data, missing values could be missing words or sentences. We would like to emphasize that missing values in language data are typically more challenging to deal with as compared with the tabular case we considered: (1) there are strong temporal dependencies in language data, (2) the length of the language data is unfixed.
>
> As a special and simple example, our method can be extended for tabular data that contains text information (for instance, the diagnosis note from the doctor). Some of such descriptions might be missing. A language embedding model could potentially be incorporated into our proposed method to deal with the missingness of such short-length text information in certain columns. Nevertheless, this would need careful design case by case.

---

> > ### Author Response · Authors · 2023-11-20
> > **Rebuttal by Authors (2/2)**
> >
> > **Q**: *How MissDiff can handle the categorical features and nonnumerical features in Tabular Data?*
> >
> > **A**: As mentioned in the second line on page 8, we adopt the same pre/post-processing as (Kim et al., 2023; Kotelnikov et al., 2022; Zheng & Charoenphakdee, 2022) for dealing with mixed-type data. It is a good point that incorporating discrete diffusion models (D3PM, tauLDR) may achieve better performance for modeling tabular data. We believe MissDiff can be developed upon D3PM as a masked version.

---

> > > ### Comment · Reviewer_JZFN · 2023-11-23
> > >
> > > Thank you very much for the detailed responses. My overall rating remains.

---

> ### Public Comment · ~Junbin_Gao1 · 2023-11-27
>
> How do you input the missing x^{obs} into the score network whose input dimension is fixed, or taking 0 values at missing position?   Thanks

---

### Official Review · Reviewer_bE5M · 2023-10-31

**Soundness:** 2 fair
**Presentation:** 2 fair
**Contribution:** 3 good
**Rating:** 5
**Confidence:** 3

**Summary:**

The authors present a new diffusion-based generative framework, MissDiff, specifically designed for learning from incomplete data and generating synthetic complete data. They highlight the common issue of incomplete data in various real-world applications, such as healthcare and finance, particularly when dealing with tabular datasets. The authors illustrate the drawbacks of existing two-stage inference frameworks, noting they are either biased or computationally taxing.

The proposed solution, MissDiff, is presented as a unified and computationally friendly framework. It models the score of complete data distribution by denoising score matching on data with missing values. The authors provide a theoretical justification for MissDiff's effectiveness and stress that the proposed training objective serves as an upper bound for the negative likelihood of observed data.

In the case of incomplete training data, MissDiff can be employed for synthetic data generation and missing value imputations based on the learned generative model. The authors conclude by stating that extensive experiments on imputation tasks and generation tasks show that MissDiff outperforms existing state-of-the-art approaches on multiple tabular datasets.

**Strengths:**

- Good coverage of existing methods
- Certainly a practically relevant problem to address.

**Weaknesses:**

- The technical description is difficult to follow, e.g. in section 2.2. or 3.2. I think it can be well understood by someone who is already familiar with the material but it doesn't do a great job building this up for the less versed reader.
- 1.1. "evidential low bound", I think you mean evidence lower bound.
- How do you tune hyperparameters, especially for competing methods? I think it's hard to compare against baselines without this.
- Claiming that existing approaches are either biased or expensive seems very general.

## Stylistic critique
- The abstract is a bit vague, e.g. "beyond missing value imputation" - not sure what that means.
- Intro: "It is known that ..." remove. Just state the claim.
- Footnote 3, you mean to say you "use them interchangeably".

**Questions:**

- I don't think that work like the VAEs for missing data can be described as "generate-than-impute". In (at least some) of these models, there is no separate imputation step, but they learn a joint density marginalizing over missing value. I may be wrong. Can you clarify?

---

> ### Author Response · Authors · 2023-11-20
> **Rebuttal by Authors**
>
> We thank the reviewer for the comments, and we appreciate the time you spent on the paper. Below we address the concerns and comments that you have provided.
>
> **Q**: *How do you tune hyperparameters, especially for competing methods?*
>
> **A**: The hyperparameters of our method can be found in Appendix B.4 (revised version updated on openreview). In short, the optimizer, learning rate, batch size, and training epochs are the same as Zheng & Charoenphakdee (2022) for fair comparison. The hyperparameters of other baseline methods can also be found in Appendix B.4.
>
> **Q**: *I don't think that work like the VAEs for missing data can be described as "generate-than-impute". In (at least some) of these models, there is no separate imputation step, but they learn a joint density marginalizing over missing value.*
>
> **A**: Thank you very much for your question. We clarify this point below. VAE-based methods (especially Mattei Frellsen (2019)) maximize the likelihood of the observed value to train an encoder projecting $x^{\text{obs}}$ to feature $z$ and a decoder projecting feature $z$ to $x^{\text{obs}}$. For the imputation tasks, they learn the distribution $p(x^{\text{missing}}|x^{\text{obs}})$ by marginalizing over latent variable $z$. However, the previous VAE-based method cannot easily deal with the generation tasks without adopting "generate-then-impute" framework. The key reason is they model the distribution $p(x^{\text{obs}}|z)$ by a student $t$ distribution with location, scale, and degrees of freedom outputted by the decoder, which has limited representation power for the real distribution. We believe directly sampling from $p(x^{\text{obs}}|z)$ for generating tasks will have poor performance. Therefore, a practical solution for generating is first generating a set of missing data and then adopting VAE-based methods to impute them.
>
> **Q**: *Claiming that existing approaches are either biased or expensive seems very general.*
>
> **A**: We provide a detailed explanation of this claim in Section 3.1 and Section 1.1.
>
> **Q**: *The technical description is difficult to follow; typos; Stylistic critique*
>
> **A**: Thank you very much for your suggestion. The typos are carefully checked and corrected. To improve the readability of Section 2.2 and 3.2, we have added specific references to the main technical equations, including the forward and backward process, the score matching objective, and variance preserving SDE.

---

> > ### Comment · Reviewer_bE5M · 2023-11-20
> > **Thank you**
> >
> > Thank you for the kind response. My overall rating remains.

---

### Official Review · Reviewer_EFAN · 2023-11-01

**Soundness:** 2 fair
**Presentation:** 2 fair
**Contribution:** 2 fair
**Rating:** 6
**Confidence:** 2

**Summary:**

This paper studies the diffusion model in the presence of missing data. The basic idea is to compute the score matching objective on observed entries. The authors proved that the solution is the true score function and this objective upper bounds the negative log-likelihood. Experiments are conducted on tabular data imputation and generating new data.

**Strengths:**

This paper proposes a new method to train a diffusion model on missing data. The training objective is simple and the authors provide theoretical guarantees.

**Weaknesses:**

1. There are several inaccurate claims about previous literature and related work, which decrease the validation of the paper.
- In Section 1.1, why did the authors claim that Mattei & Frellsen (2019); Ipsen et al. (2020b) train multiple decoders? From my understanding, they proposed multiple imputation via sampling multiple latent factors, rather than training multiple decoders.
- Ipsen et al. (2020a) was published in ICLR 2022.
- In the last line of Page 3, the authors said the references require MCAR assumption. However, I suppose most of them assumed the MAR case. Moreover, one of the contributions of Li et al. (2019) is to deal with NMAR data.
2. The authors did not mention what kind of missing mechanism was used in Section 4.1. Moreover, it seems that they only compare with baselines under one missing mechanism in this section.
3. Why did not the authors compare with CSDI_T in Section 4.2? Why are the results of STaSy not included in Table 4?

**Questions:**

While Tashiro et al. (2021) adopted conditional score matching, the work goes back to unconditional score matching. Can the authors provide more discussions about these two approaches and what are the advantages of using unconditional scores?

---

> ### Author Response · Authors · 2023-11-20
> **Rebuttal by Authors**
>
> We thank the reviewer for the comments, and we appreciate the time you spent on the paper. Below we address the concerns and comments that you have provided.
>
> **Q**: *Why did the authors claim that Mattei & Frellsen (2019); Ipsen et al. (2020b) train multiple decoders?*
>
> **A**: Thank you very much for pointing this out. We agree that Mattei & Frellsen (2019); Ipsen et al. (2020b) only train one decoder and we revise the argument in Section 1.1 accordingly in the revised paper (updated on openreview). More specifically, different from Nazabal et al. (2018); Ma et al. (2020) that training a different VAE independently to each data dimension, Mattei & Frellsen (2019); Ipsen et al. (2020b) only need to train one decoder and sample multiple latent factors from the variational distribution.
> However, these two works model the distribution $p(x^{\text{obs}}|z)$ by a student $t$ distribution with location, scale, and degrees of freedom outputted by the decoder, which has limited representation power for directly generating complete new samples. Therefore, their methods still need a two-stage inference framework for generating complete new samples, i.e., generating the samples containing missing values and adopting the proposed single imputation or multiple imputation methods to obtain complete data.
>
> **Q**: *The authors said the references require MCAR assumption. However, I suppose most of them assumed the MAR case. Moreover, one of the contributions of Li et al. (2019) is to deal with NMAR data.*
>
> **A**: Thank you very much for your comments. We agree that the architecture in Li et al. (2019) can be easily extended to MAR and MNAR cases. However, the theoretical guarantees only held for the MCAR case, which is the same as Yoon et al. (2018a). In Li & Marlin (2020), the authors assume MCAR but can still be unbiased if the missing mechanism is MAR. In Ipsen et al. (2022) and Mattei & Frellsen (2019), the authors require MAR assumptions. We have modified the last sentence on page 3 to M(C)AR accordingly.
>
> **Q**: *The authors did not mention what kind of missing mechanism was used in Section 4.1. Moreover, it seems that they only compare with baselines under one missing mechanism in this section.*
>
> **A**: We follow the same experimental setup as Zheng & Charoenphakdee (2022) for the imputation tasks. The missing mechanism is MCAR and the missing ratio is 0.2. We clarify the experimental setup in Appendix B.1. Due to the time limit, we were unable to finish the comparison with six baseline methods on six datasets under different missing mechanisms. We believe the comparison under the same setting with previous work can demonstrate the effectiveness of MissDiff for imputation tasks.
>
> **Q**: *Why did not the authors compare with CSDI\_T in Section 4.2? Why are the results of STaSy not included in Table 4?*
>
> **A**: Thank you very much for your suggestion. We have added the comparison with CSDI\_T in Table 2, 4, 7, 9 and STaSy-delete and STaSy-mean in Table 4, 9. MissDiff outperforms CSDI\_T on 8 out of 10 results and outperforms STaSy-delete and STaSy-mean by a large margin.
>
> **Q**: *While Tashiro et al. (2021) adopted conditional score matching, the work goes back to unconditional score matching. Can the authors provide more discussions about these two approaches and what are the advantages of using unconditional scores?*
>
> **A**: Thank you very much for raising this important question. First of all, there are various conditional scores (depending on which information is conditioned on), making them difficult to learn and analyze. In Tashiro et al. (2021) and its tabular variant CSDI\_T, there were no theoretical guarantees on whether the learned conditional score satisfied the optimality condition. In our work, we provide theoretical guarantees on the unconditional score learning from incomplete data. Secondly, conditional score matching performs better in time series imputation tasks than unconditional score matching, which is not necessarily the case for tabular data. We believe that is because there may exist some complex or irregular dependencies between different columns in tabular data, e.g., some features might be redundant (uninformative). Therefore, MissDiff can achieve better performance than CSDI\_T in the imputation tasks and generation tasks.

---

> > ### Comment · Reviewer_EFAN · 2023-11-21
> >
> > Thanks for the response. I would like to keep my score.

---

### Official Review · Reviewer_JCWu · 2023-11-10

**Soundness:** 3 good
**Presentation:** 3 good
**Contribution:** 3 good
**Rating:** 6
**Confidence:** 4

**Summary:**

The authors of the paper developed MissDiff, a diffusion-based method for imputing incomplete tabular data under various missingness scenarios (MCAR, MAR, MNAR) and generating data. MissDiff most appealing feature is that it imputes missing values learns the (estimated) generative distribution (i) at a unified (i.e., not multiple-stage) manner, and (b) without the computational burden of training additional networks, as opposed to the corresponding state-ot-the-art methods. The authors performed experiments comparing their method to others both with respect to data imputation and generation tasks across several datasets where, in the majority of the experiments, their method outperfomed.

**Strengths:**

--method handles tabular data;
--method imputes and generates data in without needing to train extra networks or diving the task into imputing and generating;
--both theoretical and experimental justification is presented;
--method seems to perform well under various missingness scenarios;

**Weaknesses:**

-- inconsistency of word "complete" : page 3, paragraph 2.1 : "complete d-dimensional data ... m=(m_1, ...,m_d) in {0,1}^d",  page 4, "framework for learning on incomplete data"
-- more metrics need to be presented at the experiments section, eg RMSE is not necessarily representative in high-dimensional cases were the distribution is complex-multimodal;
-- typos

**Questions:**

-- at fidelity testing (section 4.3, Figure 1) it seems that the method's performance improves as missingness rate increases in (0.1-0.6) and then decreases in the row- and column- missing setups: could you provide some insights on why this happens (also could you also add the case where missingness rate is 90% in the first case) ?
-- at table 2 it seems that MissDiff and Diff-Mean seem to have similar performance -- could you comment on that and explore more datasets?
-- could you compare your method with VAE- and GAN-based methods as well (more specifically Mattei and Frelsen, Li and Marlin) ?

---

> ### Author Response · Authors · 2023-11-20
> **Rebuttal by Authors (1/2)**
>
> We thank the reviewer for the comments and suggestions. We appreciate the time you spent on the paper. Below we address the concerns and comments that you have provided.
>
> **Q**: *Inconsistency of word "complete".*
>
> **A**: Thank you for your comments. Our proposed framework is learning from incomplete data and generating complete data. Therefore, in Section 2.1, we first define the complete (unobserved) data distribution $p_0(x)$ and then define the incomplete (observed) data samples $S^{\text {obs}}$. We have made this more clear in the revision (updated on openreview).
>
> **Q**: *More metrics need to be presented at the experiments section, eg RMSE is not necessarily representative in high-dimensional cases were the distribution is complex-multimodal.*
>
> **A**: Thank you for your good suggestion. For the generation task, we adopt two types of criteria, fidelity and utility, to evaluate the quality of the synthetic data generated. For the imputation task, we adopted RMSE in order to compare with previous works that evaluated the imputation performance mainly by RMSE.
>
> **Q**: *Could you provide some insights on why  the method's performance improves as missingness rate increases in (0.1-0.6)?*
>
> **A**: For the slight performance improvement with missing rate increase in (0.1-0.6), it is the authors' conjecture that this is a phenomenon due to the unique structure of certain tabular datasets. For this simulated Bayesian network dataset, the dependencies between different columns are demonstrated in Figure 2: some features might be uninformative, for instance, variables C1, C2, and D1 are all uninformative to the value of D3, given that D2 is observed. This implies that for some rows with missing C1, C2, and D3 values, the model still has enough information to learn the full dependence between variables D3 and D2. Moreover, the model can potentially learn the distribution of D3|D2 better in such cases since other redundant variables are excluded. We conjecture this is the reason why we see a little improvement in the performance.
>
> Nevertheless, we would like to emphasize that the improvement in performance is very small, after considering the observation noise, the authors believe it is more appropriate to conclude that the performance of the proposed method remains more ${\it stable}$ (as compared with other baselines) for missing rates in 0.1-0.6. Moreover, the performance starts to decrease when we increase the missing rate to 0.8, since in such case, we only have one variable left in each row and thus it is reasonable to expect worse performance.
>
> **Q**: *Could you add the case where missingness rate is 90\% in the first case?*
>
> **A**: As mentioned in Appendix B.1 (which is Appendix B.2 in the revised version), there are only five variables (columns) in the data generated by Bayesian Network (three categorical variables and two continuous variables). Therefore, in the row missing mechanism, we only have the missing ratio is [0.2,0.4,0.6,0.8]. For the column missing or the independent missing mechanisms, we set the missing ratio to be [0.1,0.2,0.3,0.4,0.5,0.6,0.7,0.8,0.9].

---

> > ### Author Response · Authors · 2023-11-20
> > **Rebuttal by Authors (2/2)**
> >
> > **Q**: *At table 2 it seems that MissDiff and Diff-Mean seem to have similar performance -- could you comment on that and explore more datasets?*
> >
> > **A**: Table 2 demonstrates the experimental result for the Census dataset, under a relatively simpler setting (thus an easier task) with missing ratio 0.2 and the missing mechanism being MCAR. When changing the missing mechanisms to MAR and NMAR, MissDiff performs much better than Diff-mean. We also note that MissDiff performs better for the more complicated MIMIC4ED dataset in Table 3, which contains more features (5 times than Census) and training samples (22 times than Census).
> >
> > **Q**: *Could you compare your method with VAE- and GAN-based methods as well (more specifically Mattei and Frelsen, Li and Marlin) ?*
> >
> > **A**: The key reason that we have not involved the comparison with VAE-based methods in generation tasks is that the original implementation of their method cannot be directly adopted for generation tasks. For Mattei and Frelsen, they model $p(x^{\text{obs}}|z)$ by a student $t$ distribution, whose representation power could be limited for modeling the real distribution. To obtain better performance, their methods need to further adopt the "generate-then-impute" framework, which can be performed in multiple ways. For instance, we can modify the architecture of the decoder for generating $x^{\text{obs}}$ and adopt their methods for imputation; or we can adopt the framework proposed by [1] to obtain the missing value to be imputed and adopt their methods for imputation. We believe these modifications could be helpful under careful designs, are an interesting topic to explore as future work, and might be out of the scope of this paper. For Li and Marlin, it can be adopted for generating time series data. Additional modifications should be made for generating tabular data.
> >
> > [1]: Neves, D., Alves, J., Naik, M.G., Proença, A.J., & Prasser, F. (2022). From Missing Data Imputation to Data Generation. J. Comput. Sci., 61, 101640.

---

> > > ### Comment · Reviewer_JCWu · 2023-11-23
> > > **Final decision**
> > >
> > > We thank the authors for their response
> > > Given that they did not provide empirical results, as requested, against competing methods, the final score will be 6: marginally above the acceptance threshold as we recognize the potentials of this work but there is more to be explored in order to assess its performance compared to existing methods.

---

> > > > ### Author Response · Authors · 2023-11-23
> > > > **Rebuttal by Authors**
> > > >
> > > > We thank the reviewer for the comments and suggestions. We appreciate the time you spent on the paper.
> > > >
> > > > In Table 1, We compared with VAE-based (Mattei and Frelsen) and GAN-based methods on imputation tasks, and MissDiff outperforms them by a large margin. In generation tasks, (Mattei and Frelsen) and (Li and Marlin) have their own limitations that cannot be directly used for generating tabular data or have limited representation power for generation by student t distribution with location, scale, and degrees of freedom outputted by the decoder (which can be found in the second response in Rebuttal by Authors (2/2)). Therefore, careful design should be made to their methods for generation.
> > > >
> > > > To address your further concerns, we first provide the experimental results that directly use the student t distribution learned from MIWAE for generation ("MIWAE" in the following table). We find the utility performance and the fidelity performance as bad as we anticipated. Then, we modify MIWAE by adopting the framework similar to [1], i.e., generate the missing value to be imputed and adopt MIWAE for imputation. MissDiff still achieves much better performance than MIWAE on generation tasks.
> > > >
> > > > |    | MissDiff  | Diff-delete  |  Diff-mean | STaSy-delete  | STaSy-mean  |CSDI\_T |MIWAE | MIWAE (modified)|
> > > > |---|---|---|---|---|---|---|---|---|
> > > > |  Utility evaluation | **79.48%**  | - | 78.45%  | - | 70.79%  | 79.15% | 23.7%  |  72.11% |
> > > > |  Fidelity evaluation | **80.59%**  |  - | 76.92%  | -  | 56.75%  |  77.60% | 59.11%  |  67.14% |

---

### Meta-Review · Area_Chair_p6XQ · 2023-12-12

**Metareview:**

The paper introduces MissDiff, a novel approach to learning diffusion-based models from incomplete tabular data and imputing missing values. The empirical evaluation shows that MissDiff generally outperforms existing state-of-the-art methods in data imputation and generation tasks across multiple datasets. However, parts of the paper are difficult to follow, and the authors have some substantial misunderstandings regarding existing work, which makes it hard to follow their argumentation around the two existing paradigms in learning generative models from incomplete data and how the proposed method unifies those paradigms. Reviewers found that the proposed method to train a diffusion model on incomplete data is novel, the training objective is theoretically justified, and the proposed method empirically performs well under various missingness scenarios. However, the reviewers raised issues regarding the presentation in the paper as they found that the technical description is difficult to follow and that there are inaccurate claims about previous literature and related work. They also raised issues regarding the metrics used in the empirical evaluation.

**Justification For Why Not Higher Score:**

While the paper has many strengths, the weaknesses, particularly the notable issues with clarity and inaccurate claims about previous work, place the paper just below the acceptance threshold.

**Justification For Why Not Lower Score:**

N/A

---

### Decision · Program_Chairs · 2024-01-16

Reject